# Modulation of lytic molecules restrain serial killing in γδ T lymphocytes

Patrick A. Sandoz [1,12] ✉, Kyra Kuhnigk [2,12], Edina K. Szabo[3,4], Sarah Thunberg [1], Elina Erikson[1], Niklas Sandström [1], Quentin Verron [1], Andreas Brech[5,6,7], Carsten Watzl [8], Arnika K. Wagner[2], Evren Alici[2], Karl-Johan Malmberg [2,3,4], Michael Uhlin [9,10] & Björn Önfelt [1,2,11] ✉

γδ T cells play a pivotal role in protection against various types of infections and tumours, from early childhood on and throughout life. They consist of several subsets characterised by adaptive and innate-like functions, with Vγ9Vδ2 being the largest subset in human peripheral blood. Although these cells show signs of cytotoxicity, their *modus operandi* remains poorly understood. Here we explore, using live single-cell imaging, the cytotoxic functions of γδ T cells upon interactions with tumour target cells with high temporal and spatial resolution. While γδ T cell killing is dominated by degranulation, the availability of lytic molecules appears tightly regulated in time and space. In particular, the limited co-occurrence of granzyme B and perforin restrains serial killing of tumour cells by γδ T cells. Thus, our data provide new insights into the cytotoxic arsenal and functions of γδ T cells, which may guide the development of more efficient γδ T cell based adoptive immunotherapies.

Unlike conventional T cells that bind short peptides presented by HLA-molecules, γδ T cells can recognise and exert their effector functions in response to a wider variety of molecular antigens, as reviewed in[1,2]. Among several different types of human γδ T cells, Vγ9Vδ2 is the largest subset in peripheral blood. Their T cell receptors (TCR) can sense the concentration increase of phosphoantigen metabolites, such as endogenous isopentenyl pyrophosphate (IPP) of the mevalonate pathway or microbial 4-hydroxy-3-methyl-but-2-enyl-pyrophosphate (HMBPP) which derives from pathogenic bacteria such as *Mycobacterium tuberculosis* or from malaria parasites. These phosphoantigens bind the intracellular B30.2 domain of the butyrophilin (BTN) receptor isoform −3A1, inducing a conformational change[3–5]. The signal is transduced to the γδTCR via a partly unresolved mechanism that involves interactions play with at least BTN2A1 in a complex with BTN3A1[6,7].

Due to the favourable prognostic outcome for patients with increased levels of tumour-resident γδ T cells[8,9] and the lack of MHC restriction in γδ T cell recognition, there is a growing interest in using these cells for off-the-shelf adoptive cell therapy against cancer[10,11]. In tumour cells, malfunctioning of the mevalonate pathway leads to high levels of phosphoantigens, which is recognised by Vγ9Vδ2 T cells[12]. Similarly, Vγ9Vδ2 T cells expand in vivo and ex vivo in response to zoledronate, a drug used in various bone diseases that inhibits the farnesyl pyrophosphate synthase, an enzyme acting downstream of IPP in the mevalonate pathway[13–15]. Phosphoantigen-driven expansion of γδ T cells, which rapidly provides high amounts of enriched γδ T cells, has therefore been used for immunotherapy in clinical trials against various types of cancer[16–18] but the outcome and benefits remain debated[19,20]. For example, some γδ T cell subpopulations play a

[1]Department of Applied Physics, Science for Life Laboratory, KTH Royal Institute of Technology, Stockholm, Sweden. [2]Department of Medicine Huddinge, Karolinska Institutet, Karolinska University Hospital, Stockholm, Sweden. [3]Precision Immunotherapy Alliance, University of Oslo, Oslo, Norway. [4]Department of Cancer Immunology, Institute for Cancer Research, Oslo University Hospital, Oslo, Norway. [5]Cancell, Centre for Cancer Cell Reprogramming, Department for Clinical Medicine, University of Oslo, Oslo, Norway. [6]Department of Biosciences, University of Oslo, Oslo, Norway. [7]Department of Molecular Cell Biology, Institute for Cancer Research, Oslo University, Oslo, Norway. [8]Department for Immunology, Leibniz Research Centre for Working Environment and Human Factors, TU Dortmund, Dortmund, Germany. [9]CLINTEC, Karolinska Institutet, Stockholm, Sweden. [10]Department of Clinical Immunology and Transfusion Medicine, Karolinska University Hospital, Stockholm, Sweden. [11]Department of Microbiology, Tumour and Cell Biology, Karolinska Institutet, Stockholm, Sweden. [12]These authors contributed equally: Patrick A. Sandoz, Kyra Kuhnigk. ✉e-mail: psandoz@kth.se; onfelt@kth.se

regulatory role in the tumour microenvironment, such as the IL-17 producing γδ T cells that exhibit protumour effects[21]. After allogeneic hematopoietic cell transplantation, frequencies of the γδ T cell subsets in the stem cell graft influence the treatment outcome as higher proportions of CD8+ γδ T cells are associated with increased incidence of acute graft versus host disease[22].

Phenotypically, γδ T cells show signatures of both innate and adaptive immunity[23]. The Vδ2+ compartment is known to increase in early life and consists of a Vγ9+ fraction with a semi-invariant TCR repertoire that recognises phosphoantigens[24] while the Vγ9- subset display adaptive features[25]. Single-cell RNA sequencing detected populations of Vδ1+ and Vδ2 + T cells with mature, cytotoxic phenotypes with innate features closely resembling those of natural killer (NK) cells[26,27]. In NK effector cells, cytotoxicity is induced through lytic vesicles which fuse with the effector plasma membrane. This releases cytotoxic compounds, like granzymes, and the pore-forming protein perforin at the effector-target contact site, called the immune synapse, causing target cell death through apoptosis[28,29]. Alternative apoptosis-inducing mechanisms have been described, such as the engagement of death receptors (FAS and TRAIL) with their ligands[30]. In γδ T cells, although cytokine secretion and cytolytic activity have been reported previously[11,31], the dynamic aspects of their immune surveillance and their killing mechanisms remain largely unknown. In particular, high-resolution microscopy of γδ T cells has been overlooked and γδ T cells effector-behaviours have not been characterised in detail by live-cell imaging. This lack of fundamental knowledge about γδ T cells effector mechanisms currently undermine the development of more effective cell-based therapy. This is particularly true for the Vγ9Vδ2 T cell subset which does not exist in rodents and is therefore more difficult to study in vivo.

Our present work aims to provide a temporal framework of the phenotypic profiles and cytotoxic functions of Vγ9Vδ2 T cells during their phosphoantigen-induced activation and expansion. Vγ9Vδ2 T cells show twenty-fold upregulation of granzyme B in a narrow time window of a few days after phosphoantigen stimulation. Microwell-based live-cell imaging of Vγ9Vδ2 T cells interacting with tumour cells expressing fluorescent reporters for enzymatic activity allows us to characterise killing mechanisms and dynamics at the single-cell level. Only a minority of the Vγ9Vδ2 T cells is able to kill several target cells in sequence. While such serial killers predominantly use granzyme B-mediated mechanisms, cells that kill fewer target cells often use death receptor signalling or a so far unidentified mechanism. Super-resolution fluorescence and electron microscopy imaging show that the spatial distribution of granzyme B and perforin in the lytic granules is imbalanced, consistent with perforin being limiting for efficient killing during early phases of the expansion. Donors of Vγ9Vδ2 T cells with the highest co-occurrence of granzyme B and perforin show the most efficient killing of tumour cells. Together, our single-cell imaging represents an important first step to fill the vast knowledge gap existing for γδ T cells and especially provides insights into the cytolytic mechanisms used by γδ T cells. This will support the development of efficient off-the-shelf products for cellular immunotherapies.

## Results

### Granzyme B is upregulated during early phases of phosphoantigen-triggered Vγ9Vδ2 T cell expansion

The dynamics of Vγ9Vδ2 T cell activation and expansion triggered by phosphoantigens have been unclear. Therefore, we explored the response of peripheral blood mononuclear cells (PBMC)-derived Vγ9Vδ2 T cells stimulated by zoledronate and interleukin-2 (IL2) over 14 days. This method mainly generates Vγ9Vδ2 effector memory T cells (CD45RO+CD27−) with cytotoxic properties[13–15]. Using flow cytometry, Vγ9Vδ2 T cell phenotypes were assessed at multiple time points during the 14 days of culture (Fig. 1a). The majority of Vγ9Vδ2 T cells differentiated into effector memory cells within the first 3 days (Fig. 1a

Supplementary Fig. 1a). Monitoring cell proliferation showed that most of the expansion occurred after the third day (Fig. 1b, c), and the proportion of Vγ9Vδ2 T cells in the cultures increased dramatically around the fourth day (Fig. 1d and Supplementary Fig. 1b) with only a minor residual fraction of Vγ9Vδ2 T cells that had not divided by then (Fig. 1c). Together, these observations suggest that phosphoantigen recognition leads to two successive phases in Vγ9Vδ2 T cells: a 3-4-day maturation followed by massive proliferation.

Cytotoxic profiles have been identified in circulating Vγ9Vδ2 T cells based on the surface expression of CD27, CD28, and CD16, including, e.g., a scarce CD27[neg]CD28[neg]CD16[pos] mature subset which showed enhanced cytotoxicity[32]. Using flow cytometry, we observed that most of the previously reported phenotypes based on these markers were lost during the early expansion phase (Fig. 1e, f). On day 3 of the expansion, the majority of Vγ9Vδ2 T cells had adopted a CD27[neg]CD28[pos]CD16[neg] phenotype (Fig. 1f, g and Supplementary Fig. 1c). The phenotypes later changed to more heterogeneous populations with partly restored CD16 expression (Fig. 1e–g and Supplementary Fig. 1c, d), as also recently observed in Vδ2 + T cells from patients, ten days after acute malaria infection[33].

These relatively rapid phenotypic changes over the first days of expansion made us investigate how the lytic arsenal of the Vγ9Vδ2 T cells was affected. As previously reported[32], resting Vγ9Vδ2 T cells (day 0) showed donor-dependent levels of granzyme B related to their CD27/CD28/CD16 profiles (Supplementary Fig. 1e, f). Nevertheless, all donors upregulated granzyme B to very high levels (GrzB[bright]) between days 3 to 5, which later returned to lower levels (GrzB[dim]) on days 6-14 (Fig. 1g, h and Supplementary Fig. 1g). In comparison to its resting level, granzyme B was upregulated ~20 fold as shown by flow cytometry (Fig. 1g, i) and ELISA (Supplementary Fig. 1h) but only when the cells were stimulated with both zoledronate and IL2 (Supplementary Fig. 1i). In line with the previous observation that monocytes activate Vγ9Vδ2 T cells in vitro[34] and during bacterial infection[35,36], autologous isolated monocytes were sufficient to drive the upregulation of granzyme B by Vγ9Vδ2 T cells on day 3 (Supplementary Fig. 1j). In contrast, the other granzymes (A,K,H,M) showed less pronounced differences (Fig. 1g and Supplementary Fig. 1k). The expression of perforin remained stable over the expansion, while granulysin gradually increased over the 14 days (Fig. 1g, i). The death receptor ligand TRAIL showed a slight increase on day 3 while FasL remained essentially constant during the expansion (Fig. 1g and Supplementary Fig. 1k).

Notably, several activating receptors, i.e., NKp30, NKp44, NKG2A, NKG2D, and DNAM1 were upregulated on day 3 and remained at similar levels over the rest of the expansion (Fig. 1g and Supplementary Fig. 1k). NKp46, on the other hand, was downregulated at day 3 and then recovered to the original level at day 14 (Fig. 1g and Supplementary Fig. 1k). Exhaustion markers such as PD1, CTLA4, and TIM3 exhibited minor expression differences over the 14 days (Fig. 1g and Supplementary Fig. 1k).

Furthermore, Vγ9Vδ2 T cell expansions stimulated by HMBPP (instead of zoledronate) and IL2 reached similar GrzB[bright] levels suggesting that granzyme B is similarly regulated by different phosphoantigen stimulations (Fig. 1j). Stronger stimuli with additional IL2 or IL15, reported to increase granzyme B levels of fully expanded cells[37], did not significantly enhance the granzyme B peak on day 3 (Supplementary Fig. 1l).

Upregulation of granzyme B and γδ T cell proliferation appeared to be sequential events; γδ T cells first upregulated granzyme B (day 3) and then went through several cell divisions (days 6 to 10, Fig. 1k and Supplementary Fig. 1m). During the proliferation phase the expression of granzyme B turned back to lower levels. To investigate if γδ T cells would upregulate granzyme B again if re-stimulated after the expansion, CellTrace-stained 14-days expanded γδ T cells were mixed with thawed autologous untreated PBMCs and re-stimulated with zoledronate and IL2 for 14 days. Although granzyme B was upregulated again after 3 days

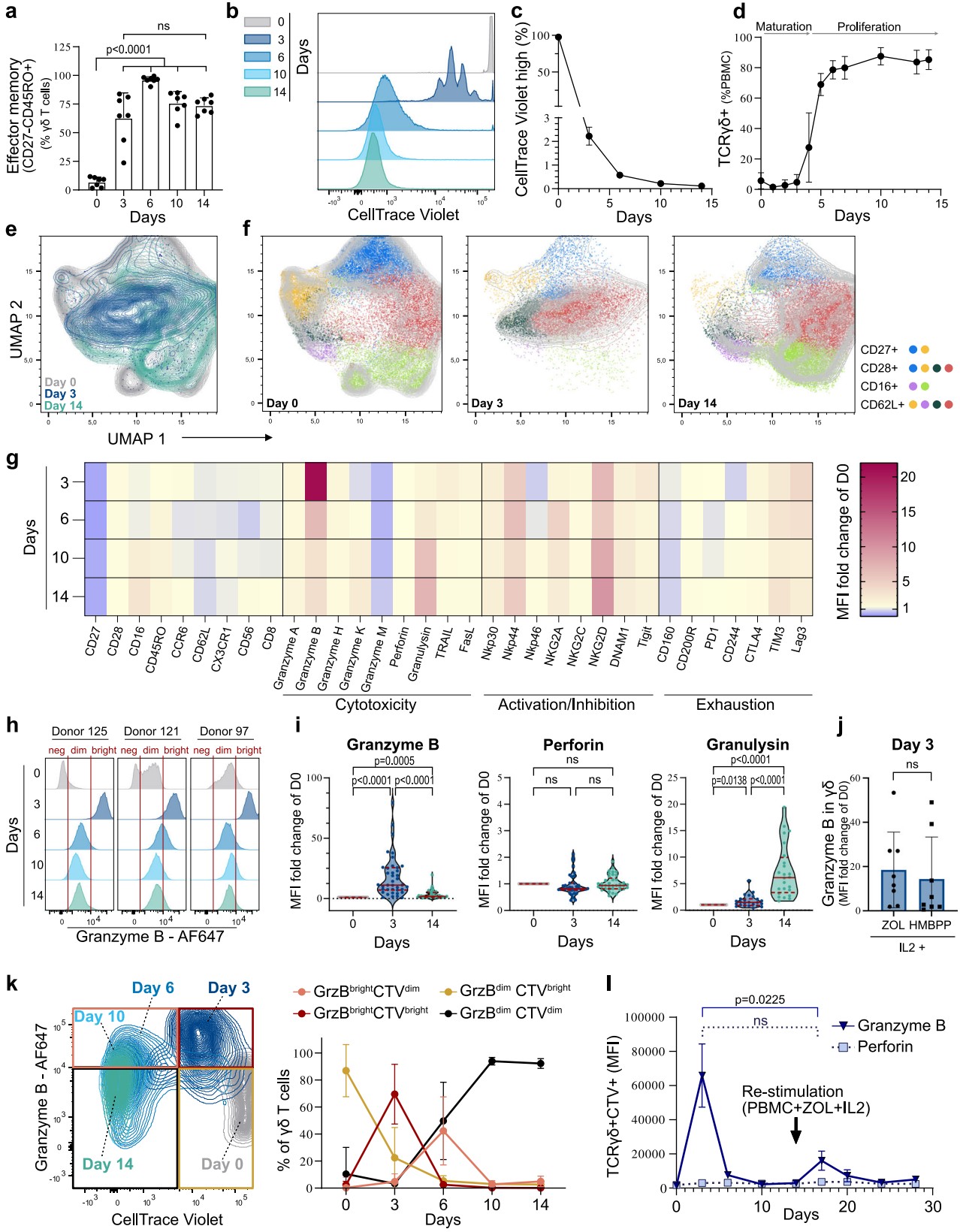

of re-stimulation, its level remained low compared to the initial peak (Fig. 1l) and was driven mainly by IL2 (Supplementary Fig. 1n). Altogether, these results show that in response to phosphoantigens, a majority of Vγ9Vδ2 T cells transiently upregulate granzyme B during maturation. This early expansion phase is accompanied by a loss of phenotypic diversity, gradually re-established during the expansion.

**Fig. 1 | Vγ9Vδ2 T cells transiently upregulate granzyme B before expansion.**
**a**–**i** Phenotypic changes and cell division of Vγ9Vδ2 T cells measured by flow cytometry in peripheral blood mononuclear cells (PBMC) stimulated with zoledronate (ZOL) and IL2 at the indicated days. **a** Proportion of γδ T cells displaying an effector memory phenotype (CD27−CD45RO+), $n = 7$ donors, ANOVA statistical comparison. **b** Proliferation of γδ T cells in PBMCs initially stained by Cell-Trace Violet (CTV). **c** Percentage of CTV-stained γδ T cells which maintained the initial CTV high signal and therefore did not divide at the different time points, $n = 3$ donors. **d** Percentage of γδ T cells during the expansion, $n = 4$ donors. **e** UMAP dimension reduction of flow cytometry measurements of γδ T cells stained for CD27, CD28, CD16, CD62L, CCR6, and NKG2A during the expansion (contour plot with outlier points), $n = 10$ donors. **f** FlowSOM clustering of the γδ T cell as six subpopulations shown on the UMAP analysis (indicative contour plots from Fig. 1e in light grey). The markers mainly driving the population dynamics are shown on the right. **g** MFI fold-change of the indicated proteins during the expansion (normalised to MFI on day 0), $n =$ between 6 (exhaustion) to 47

(granzyme B & perforin) donors. **h** Example of three donors with GrzBneg, GrzBdim and GrzBbright levels at day 0. **i** Violin plot (dots: individual donors; centre line: median; dotted lines: upper and lower quartiles) and ANOVA statistical comparison of expression fold-changes. **j** Granzyme B fold-change in expression at day 3 for PBMCs stimulated with IL2 and ZOL or IL2 and HMBPP, $n = 8$ donors. **k** Expression of granzyme B and intensity of CTV in γδ T cells during the expansion. The curves (right panel) report the proportion of high or low expressing populations gated as indicated in the example contour plot, $n = 10$ donors. **l** Expression of granzyme B and perforin during re-stimulation of 14-days expanded CTV-stained γδ T cells with ZOL and IL2 in thawed autologous unstained PBMCs for additional 14-days. Granzyme B amounts were measured by flow cytometry in the CTV+ γδ T cells, $n = 3$ donors. All applicable plots show means with standard deviations unless specifically stated otherwise. **a**, **i**–**j** one dot represents the measurement for one donor. Statistics were calculated using two-tailed ratio paired $t$-test unless stated otherwise. ns: not-significant.

## Vγ9Vδ2 T cell lytic vesicles are loaded and mature

Using anti-granzyme B labelling of sectioned cells in combination with electron microscopy, we confirmed that granzyme B was mostly localised in dense-core granules. We compared Vγ9Vδ2 T cells that had been expanded for 4 days (γδ^D4, amid peaking granzyme B expression) or 14 days (γδ^D14). For comparison, we also analysed IL2-activated NK cells (NK^D4) from the same donors (Fig. 2a). The average number of granules per cell section was similar in all three conditions, but the number of granzyme B-loaded granules was about twice as high in γδ^D4 T cells compared to γδ^D14 or NK^D4 cells, and with about twice the amount of granzyme B gold particles counted per granule in γδ^D4 T cells (Fig. 2b and Supplementary Fig. 2a, b). Granzyme B+ granules were larger in γδ^D4 T cells (Fig. 2c and Supplementary Fig. 2c) but no difference in shape was noticed (Supplementary Fig. 2d). However, there was no strong correlation between granule size and granzyme B loading (Supplementary Fig. 2e). Interestingly, γδ^D4 T cells also had more chondroitin sulfate, which is associated with serglycin, a core protein within the dense core of cytotoxic granules, (Fig. 2d, e and Supplementary Fig. 2f) but the same level of CD63, a late endosomal/lysosomal marker, (Supplementary Fig. 2g), suggesting that the granules in γδ^D4 T cells are more loaded and therefore possibly more cytotoxic.

## Granzyme B upregulation leads to more granzyme B-mediated killing in γδ^D4 T cells

Vγ9Vδ2 T cells' lytic strategies and killing dynamics have so far remained unexplored. We, therefore, aimed to study in-depth how Vγ9Vδ2 T cells attack tumour cells and how the phenotypic changes seen during the expansion (shown in Fig. 1) affected their cytotoxic functions. For this, we used an approach that we developed to study NK cell cytotoxic behaviour[38,39], where individual effector cells were monitored by time-lapse microscopy screening while interacting with tumour cells in a microwell chip (Fig. 3a). As target cells we used A498 kidney carcinoma cells expressing a dual fluorescent reporter for granzyme B- and caspase-8 to distinguish granule- and death receptor-mediated killing[40] (cell line referred to as A498^GBDR, Supplementary Fig. 3a). At steady state, the two fluorescent moieties of the reporter are expressed in the cytosol but excluded from the nucleus. Under cytotoxic attack, the corresponding reporter construct is cleaved allowing the fluorescent protein to diffuse into the nucleus (Supplementary Fig. 3b). A498 cells were selected as targets because they express both butyrophilins and death receptors (Supplementary Fig. 3c).

γδ^D4 or γδ^D14 T cells were isolated and seeded with A498^GBDR cells on a multichambered chip with 80 μm wide microwells at an effector-to-target (E:T) cell ratio of 1:1. Cytotoxicity was assessed with and without the addition of the phosphoantigen HMBPP, which once

loaded by tumour cells trigger Vγ9Vδ2 activation via the TCR. This method allowed us to assess the killing capacity, cytotoxic mechanism, and contact dynamics in detail (schematically shown in Fig. 3b). Unstimulated γδ^D4 T cells killed about 20% of the A498^GBDR cells while HMBPP stimulation increased the killing to about 80% for both γδ^D4 and γδ^D14 T cells (Fig. 3c). This was also reflected in the fraction of wells in which all targets were killed (Supplementary Fig. 3d). The spontaneous death of the A498^GBDR (observed in the wells lacking Vγ9Vδ2 T cells) remained negligible (<1%). Thus, Vγ9Vδ2 T cells showed considerable basal cytotoxicity, which increased significantly with the engagement of the TCR.

For killing, γδ^D4 T cells used granzyme B (~40% of the events) and death receptors (~15% of the events) independently of the HMBPP stimulation (Fig. 3d and Supplementary Movies 1 and 2). The remaining fraction, which included unstained or masked target cells and other putative mechanisms, is investigated further in the sections below. Compared to γδ^D4, γδ^D14 T cells used granzyme B to a lower extent (Fig. 3d) and overall killed fewer A498^GBDR using this pathway (Fig. 3e) with or without HMBPP stimulation. This difference in killing efficiency of untreated target cells between γδ^D4 and γδ^D14 T cells was confirmed using flow cytometry measuring cytotoxicity of γδ T cells co-cultured with the human B cell line Daudi (Supplementary Fig. 3e). Together, TCR-mediated recognition of phosphoantigens was a strong trigger for Vγ9Vδ2 T cells to kill, partly through granzyme B but also through other pathways. The increased fraction of granzyme B-mediated killing observed in γδ^D4 compared to γδ^D14 T cells correlates with their higher expression of granzyme B.

## γδ^D4 T cells show rapid and effective granzyme B-mediated killing

Next, we studied the timing of γδ T cell killing. As previously observed in NK cells[38], granzyme B killing occurred early in the assay (Fig. 3f, 50% of kills within 65 min) compared to death receptor (50% >4 h). γδ^D14 T cells did not kill enough untreated A498^GBDR (Fig. 3c, e) to provide a robust statistical analysis of these dynamics (e.g., only 7 out of 1083 targets were killed by granzyme B), and these data were therefore excluded from Fig. 3f–j. Upon HMBPP treatment, granzyme B-mediated killing was detected earlier in the assay for γδ^D4 compared to γδ^D14 T cells (Fig. 3f). The time interval between the fluorescent protein diffusion into the nucleus and the actual death of the target cell (phase iv-v in Fig. 3b) was shorter for γδ^D4 compared to γδ^D14 T cells, suggesting a more powerful lytic hit by γδ^D4 T cells (Fig. 3g). The time between the start of the assay until the initiation of the first lethal contact between individual T cells and target cells (phases i-ii) and the time from the beginning of individual lethal contacts to target cell death (phases ii–v) showed only minor differences between conditions (Supplementary Fig. 3f, g).

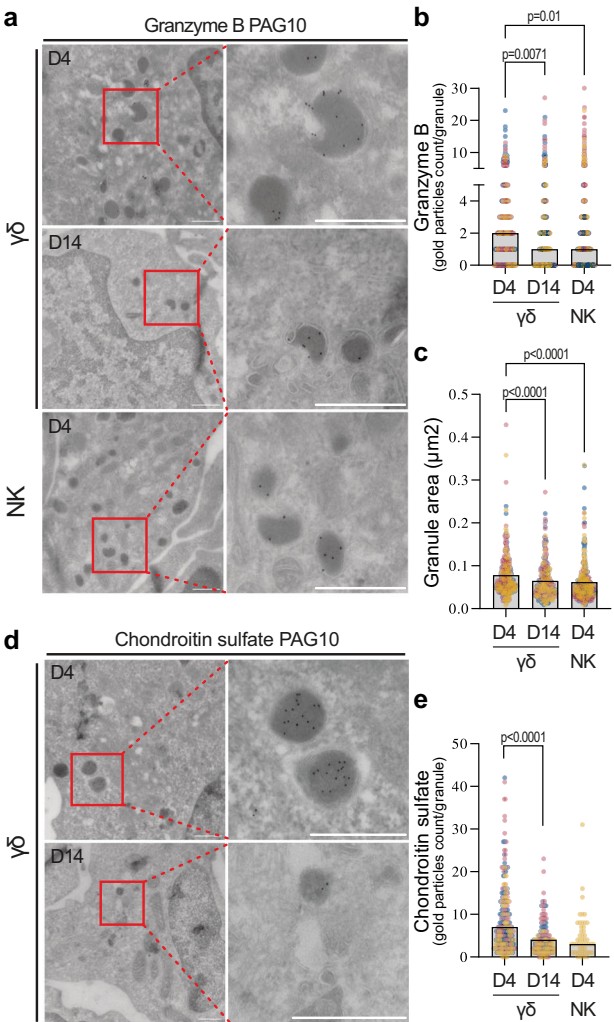

**Fig. 2 | Vγ9Vδ2 T cell lytic vesicles are mature and loaded with granzyme B.** Anti-granzyme B (**a**–**c**) or anti-chondroitin sulfate (**d**, **e**) labelling of cell sections from isolated γδD4 or γδD14 T cells or isolated IL2-activated NKD4 cells from the same donors, imaged by transmission electron microscopy. 20 cells from the cell sections were imaged for each donor. **a** Representative immuno-EM sections labelled with anti-granzyme B followed by protein A gold. Scale bars: 0.5 μm. **b** Number of gold particles (granzyme B) per granule for the different conditions, the bar plots represent the medians. **c** Quantification of the area of the dense core granules, the bar plots represent the medians (**b**, **c**, γδD4 n = 304 granules, γδD14 n = 264 granules, NKD4 n = 311 granules), the data are pooled from 3 donors. **d** Representative immuno-EM sections labelled with anti-chondroitin sulfate followed by protein A gold. Scale bars: 0.5 μm. **e** Number of gold particles (chondroitin sulfate) per granule for the different conditions, the bar plots represent the medians (γδD4 n = 240 granules, γδD14 n = 163 granules, NKD4 n = 50 granules), the data are pooled from 3 donors (γδ4, γδD14) and 1 donor (NKD4). **b**, **c**, **e** The individual dots represent one granule and they are colour-coded according to donor. All analysis by Kruskal–Wallis test.

## The unidentified killing mechanism shows intermediate dynamics

The unidentified death events included insufficiently labelled or masked targets, double positive signals in the nucleus, or complete absence of signal in the nucleus (Supplementary Fig. 3h). The average time between contact and target cells' death (phases ii–v) was in-between the times found for granzyme B and death receptor (Supplementary Fig. 3g). To assess the fraction of non-granzyme B, non-caspase-8 mediated death, we looked in more detail into the time span ii–v for individual events (Fig. 3h). The killing events could be sorted into three categories: fast (granzyme B-like) killing,

slow (death receptors-like) killing, and intermediate killing. This last category was unlike granzyme B or death receptor killing and corresponded to ~25% of the killing events (Fig. 3h and Supplementary Fig. 3i, j). It was characterised by a slow collapse of the dying target cell with no clear signal in the nucleus (Supplementary Movie 3).

It has previously been reported that lymphocytes can kill intracellular parasites by granulysin[41], and we, therefore, speculated that granulysin could be responsible for the intermediate-timed death observed here. The addition of recombinant granulysin on A498^GBDR cells showed that very high concentrations (>15 μM) are required to induce target death (Supplementary Movie 4). Interestingly, granulysin-induced target death showed slow collapse and nuclear membrane rupture consistent with our observation of the unknown mechanism in γδ T cells. It is plausible that granulysin is responsible for the intermediate killing mechanism. However, in the absence of an available granulysin-specific inhibitor, this hypothesis will need to be confirmed in a separate study.

## Granzyme-B-mediated killing mainly occurs via committed contact with the tumour cell

By tracking single γδ T cells, we could quantify the time spent in contact with target cells and characterise the type of contact. We separated "committed" from "uncommitted" contacts. In committed contacts, the effector cell stopped and spread out across the target cell for at least 2 frames (6 min), with a morphology consistent with the formation of an immune synapse[28]. In uncommitted contacts, the effector cell remained motile (Supplementary Movie 5), which could be equivalent to kinapses described for conventional T cells[28]. In untreated target cells, granzyme B killing was mainly observed in committed contacts (Fig. 3i), while killing through death receptors or the intermediate mechanism mainly occurred through uncommitted contacts. For targets treated with HMBPP, thus leading to engagement of the TCR, the fraction of committed contacts increased for all killing mechanisms (Fig. 3i). HMBPP-treatment was also accompanied by prolonged periods of attachment between γδ T cells and tumour cells (Fig. 3j), where more than half of the γδ T cells never detached from their targets during the 16-h assay. Only minor differences in type and length of contacts were observed between γδ^D4 and γδ^D14 T cells. Upon HMBPP treatment, events of fratricide (γδ T cell killing another γδ T cell) were sometimes observed in microwells with several effector cells but no targets (Supplementary Movie 6).

## Vγ9Vδ2 T cell serial killers mainly use granzyme B-mediated killing

Next, we investigated the serial killing capacity of individual γδ^D4 T cells. To give each effector cell the possibility to interact with several A498^GBDR, we performed these experiments in a microwell chip with larger wells (350 μm) where there were ~20 target cells and ~5 effector cells per well. In this setting, we recorded slightly lower overall killing (Supplementary Fig. 4a), likely due to the smaller global E:T ratio but the cytotoxic strategies and dynamics remained similar (Supplementary Fig. 4b, c and Supplementary Movie 7) as in the smaller microwells. HMBPP treatment strongly increased the number of cytotoxic γδ^D4 T cells (Fig. 4a). The majority of γδ^D4 T killers had only killed a single target cell both with and without HMBPP treatment (Fig. 4b). However, some of the γδ^D4 T cells were able to kill several target cells in sequence, and this fraction of serial killers increased by HMBPP treatment (Fig. 4b–d). The vast majority of serial killers killed their targets in an uninterrupted sequence before they finally stopped killing, although they frequently continued to interact with remaining A498^GBDR (Supplementary Fig. 4d). Very few cells alternated between killing and not killing. This suggests that γδ^D4 T cell serial killers are strongly activated and continue killing until exhausting their cytotoxic potential.

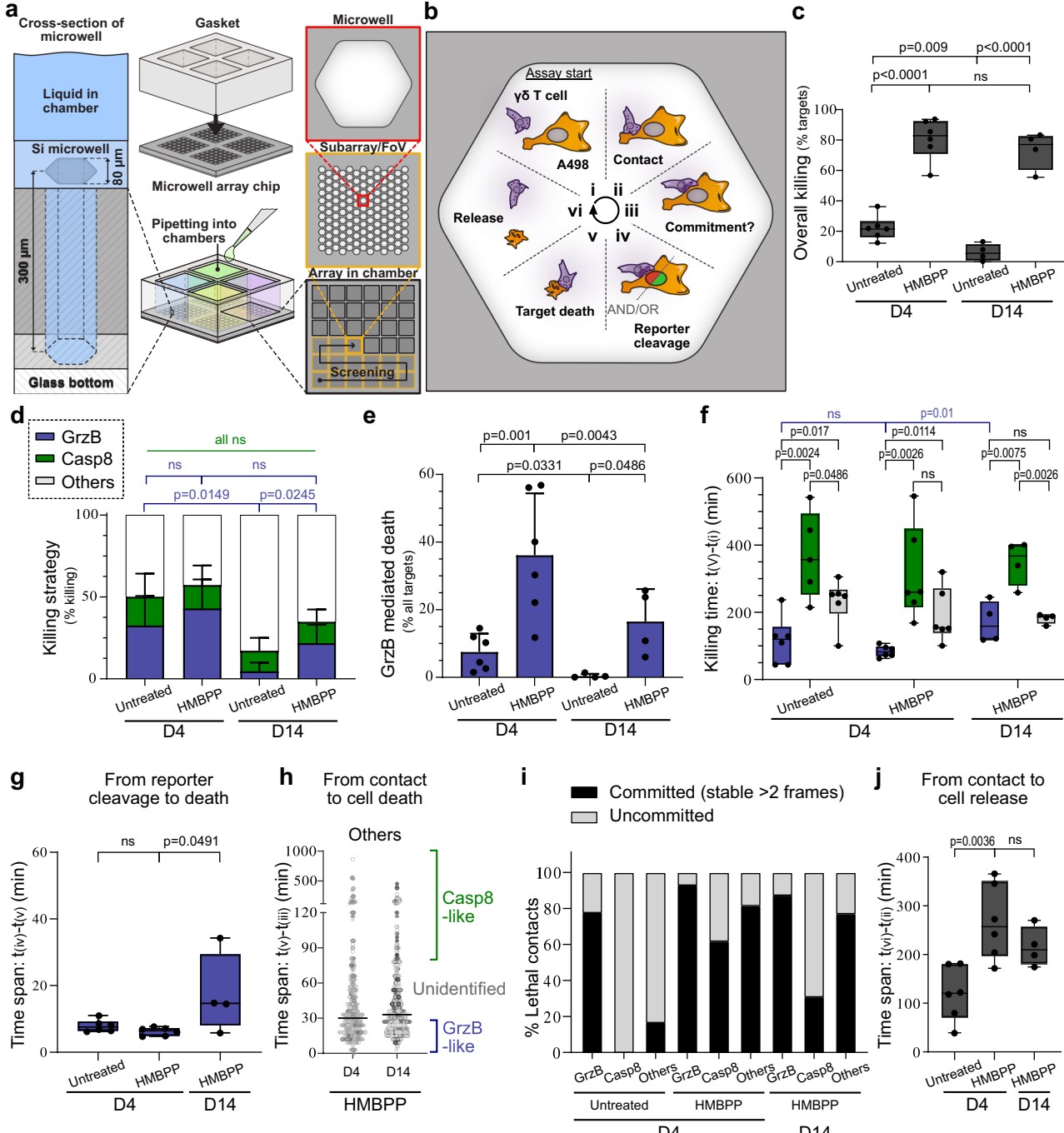

**Fig. 3 | Individual Vγ9Vδ2 T cell killing potential and cytotoxic strategies.**
**a** Scheme of the screening assay. The silicon (Si)-glass chip consisted of four separate compartments (middle) with each having an array of 300 μm deep, 80 μm wide honeycomb wells (left). The cells are confined in deep microwells (right) and imaging every 3 min enabled us to track every Vγ9Vδ2 T cell. **b** A498GBDR seeded in a microwell and expressing the two moieties of the fluorescent reporter for different killing mechanisms. Schematic of the successive phases (i−vi) and the measured parameters of the attack and killing of a A498GBDR cell by a γδ T cell. **c** Killing of the A498GBDR targets by γδD4 or γδD14 T cells +/− HMBPP, n = 6 (untreated: 156 killing events, HMBPP: 749 killing events), and n = 4 (untreated: 47 killing events, HMBPP: 595 killing events) donors, respectively. **d** Proportions of killing strategies by analysis of the reporter. **e** Proportions of granzyme B-mediated killing of all target cells by γδD4 or γδD14 T cells. **f** Time of the killing event (from the start of the assay until target death) for each mechanism. **g** Time span [iv−v] for

granzyme B-mediated death. **h** Refined classification of the killing dynamics of the other mechanisms into three categories: fast granzyme B-like (within 75% of the fastest granzyme B-mediated killing dynamics), slow death receptors-like (within 75% of the slowest death-receptors-mediated killing dynamics); or intermediate that remains unidentified (in between fast and slow). Individual dots represent single target cell deaths and they are colour-coded according to donor. **i** Type of γδ T cell−A498GBDR contacts leading to the target's death. Classification as: committed (stable >6 min); or uncommitted (=kinapses, stable ≤ 6 min). **j** Time span [ii−vi] of γδ T cell−A498GBDR interactions leading to target's death. All the plots show means with standard deviations unless stated otherwise. **c, f−h, j** The box line shows the median, the limits represent upper and lower quartiles, and whiskers indicate the minimum and maximum values. **c, e−h, j** One dot represents the measurement for one donor. Statistics were calculated using two-tailed Student's t-test. ns: not-significant.

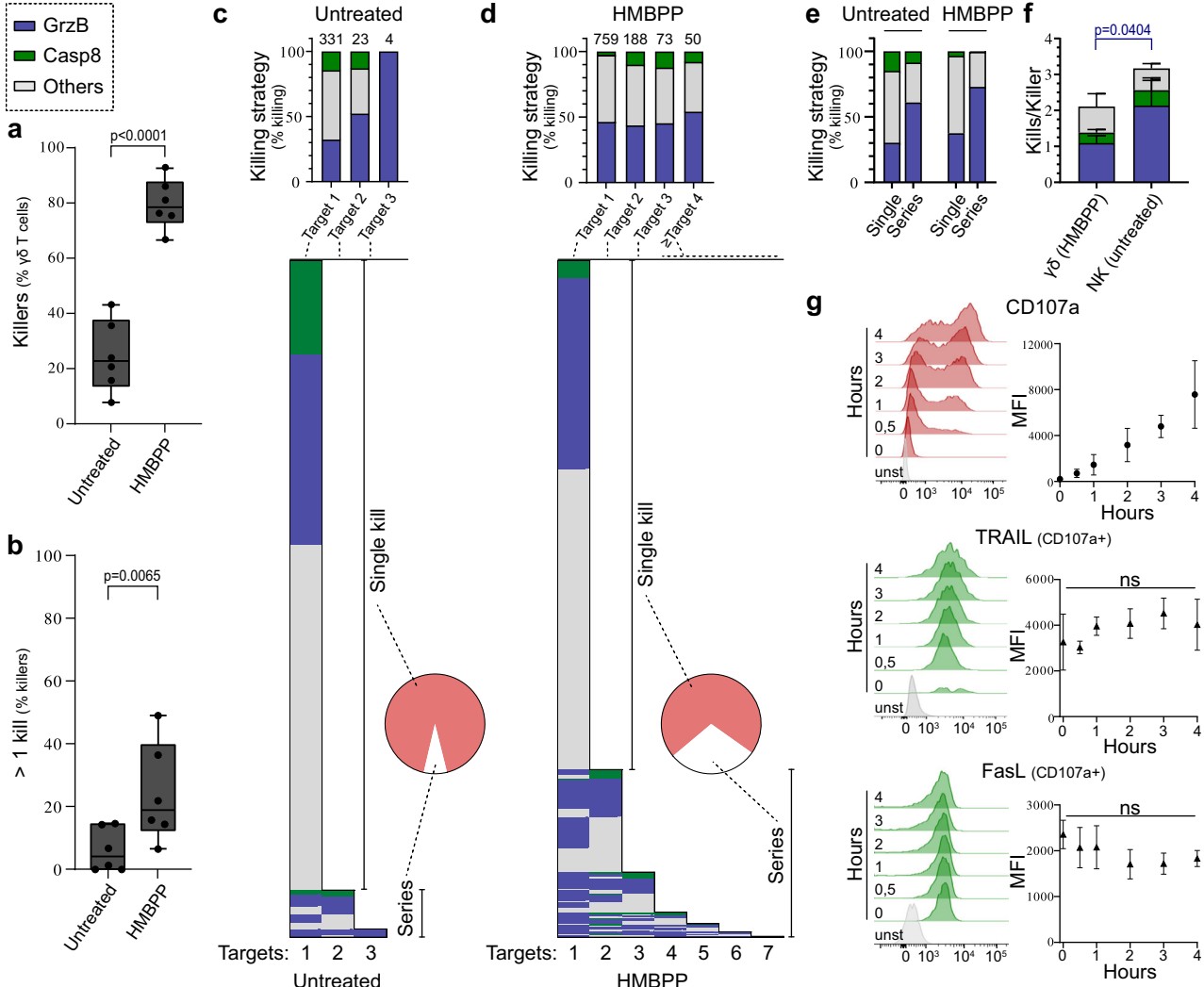

**Fig. 4 | Characterisation of Vγ9Vδ2 T cell serial killing dynamics. a–d** Measured killing of the A498GBDR targets by γδD4 T cells +/− 1.5 µM HMBPP in 350 µm-well chip during 16 h of imaging each position every 3 min. **a** Percentage of γδ T cell that kill at least one target $n = 6$ donors and **b** percentage of γδ T cell killers that killed more than one target cell, $n = 6$ donors. The box limits represent upper and lower quartiles, and whiskers indicate the minimum and maximum values. One dot represents the measurement for one donor. **c, d** Proportion of serial killing (Pie chart) and killing strategies used during serial killing aggregated for all γδD4 T cells (top bar graph) or for individual γδD4 T cells (lower diagram). Numbers above each bar of the top bar graph represent the respective number of events. Results from 6 donors were pooled. **c** Control condition and **d** HMBPP-treated cells. **e** Proportions of killing strategies comparing γδD4 T cells killing a single target cell with the first target killed from γδD4 T serial killers (data from **c, d**). **f** Number of killed target cells per γδD4 T cell (targets treated with 1.5 µm HMBPP) or NKD4 cell killers for each killing mechanism, $n = 4$ donors. **g** In-tube 4 h degranulation assay of γδD4 T cells against wild-type A498 measured by flow cytometry at the indicated time-points. Surface expression of CD95/FasL ($n = 3$ donors) and TRAIL ($n = 4$ donors) showed after gating on degranulating (CD107a+) γδ T cells. Control is unstained. All applicable plots show means with standard deviations unless specifically stated otherwise. Statistics were calculated using two-tailed paired *t*-test (**a, b, f**) and one-way ANOVA (**g**). ns=not-significant.

Comparing killing strategies used by γδD4 T cells that killed only one target with the first kill performed by serial killers revealed that serial killers used granzyme B to a larger extent (Fig. 4e). Thus, this suggests a link between serial killing and degranulation-mediated killing. However, overall the number of granzyme B-mediated kills per killer was significantly lower in γδD4 T cells compared to NKD4 cells (Fig. 4f and Supplementary Fig. 4e). Unlike NK cells[38], γδD4 T cell serial killers did not switch killing mechanism to using death receptors at the end of serial killing sequences (Fig. 4c, d). In NK cells, this switch was accompanied by surface upregulation of the death receptor FasL after killing[38]. For γδD4 T cells, no upregulation of the death receptor ligands FasL or TRAIL was observed within 4 h of co-culture (Fig. 4g). These results show that even if a large majority of γδD4 T cells were cytotoxic upon phosphoantigen stimulation, the killing potential was often limited to one kill (≥50% of the cytotoxic cells in all donors,

Supplementary Fig. 4f). The population of γδD4 T serial killers preferentially used granzyme B- rather than death receptors-mediated killing.

**Perforin availability tunes granzyme B-mediated serial killing**

To validate that granzyme B-mediated killing was occurring downstream of the TCR, we added anti-γδTCR blocking antibodies in killing assays. This abrogated the killing upon stimulation with HMBPP but did not affect the killing of untreated A498GBDR (Fig. 5a). This suggests that γδD4 T cells recognise unstimulated target cells through TCR-independent pathways (e.g., NKG2D or NCRs). Minor differences in the distribution of killing strategies were observed with and without treatment with phosphoantigens and antibody-blocking (Supplementary Fig. 5a), with only slightly increasing granzyme B-mediated killing upon TCR involvement.

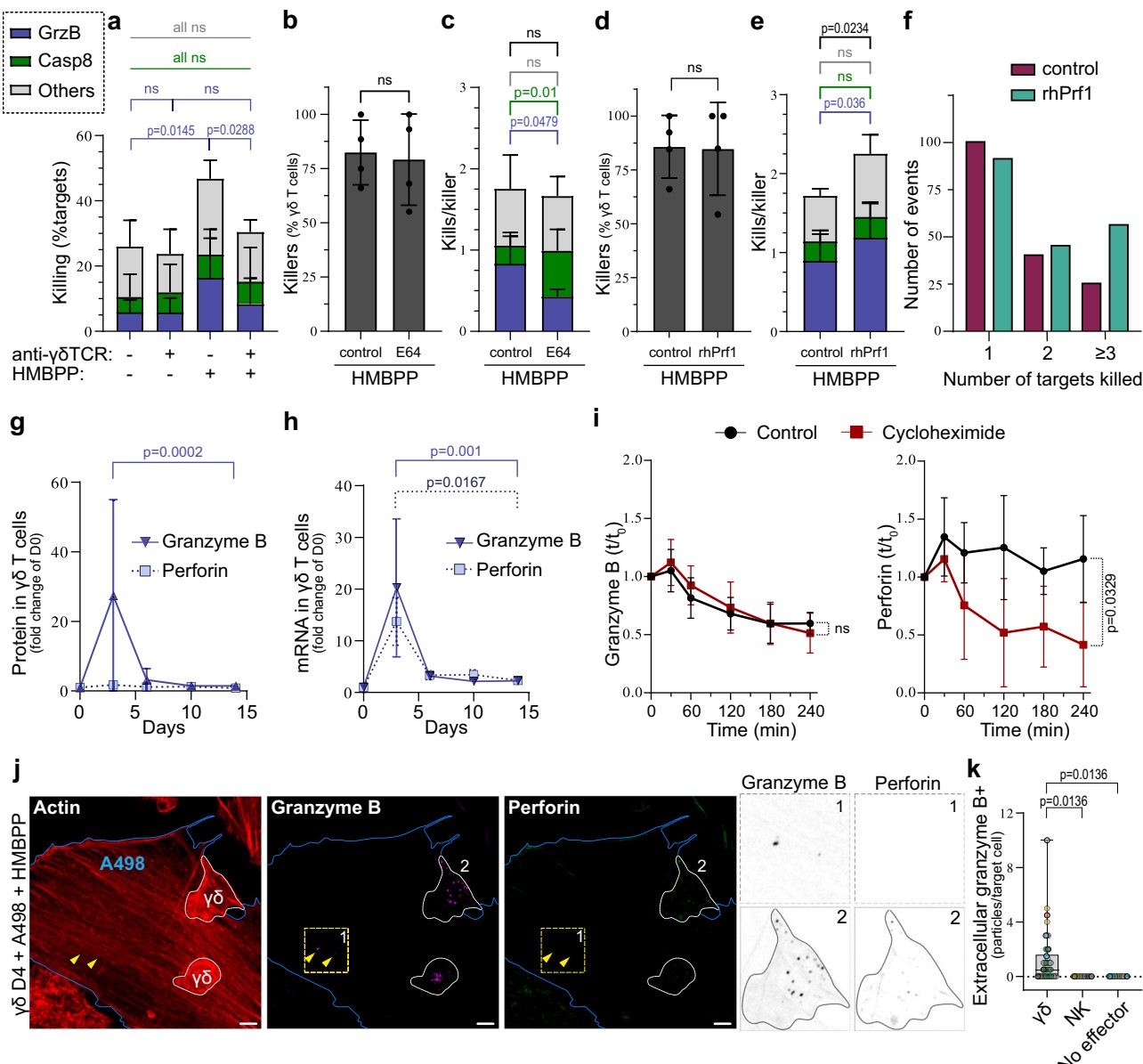

**Fig. 5 | Perforin availability is a bottleneck for γδ T cell serial killing. a** Percentage of all targets killed through different mechanisms upon anti-γδTCR blocking (or IgG antibody control) and treated or not with HMBPP, *n* = 3 donors. **b, c** On-chip killing assay with γδD4 T cells against A498GBDR treated with HMBPP in the presence or not of E64 inhibitor, *n* = 4 donors. **b** Percentage of γδ T cell killers. **c** Number of killed target cells per γδ T cell killer for each killing mechanism. **d–f** On-chip killing assay of γδD4 T cells against A498GBDR treated with HMBPP in the presence or not of recombinant human perforin (rhPrf1) in the medium, *n* = 4 donors. **d** Percentage of γδ T cell killers. **e** Number of killed target cells per γδ T cell killer for each killing mechanism. **f** Number of kills performed by individual γδ T cell killers for the four pooled donors. **g, h** PrimeFlow measurement of γδ T cell protein (**g**) and mRNA (**h**) levels for granzyme B (*n* = 6 donors) and perforin (*n* = 3 donors) at the indicated timepoints. **i** Granzyme B (left) or perforin (right) content in γδD4 T cells as measured by flow cytometry at the indicated timepoints in degranulation assays with or without cycloheximide, *n* = 6 donors. Statistical test

on the last time point. **j** Co-culture of γδD4 T cells with wild-type A498 targets with HMBPP prior immunofluorescence labelling for granzyme B, perforin and actin. Boundaries of γδ T cells and A498 cells are marked in white and blue respectively. Marked square (1) and γδ T cell (2) are zoomed in to the right. Yellow arrows point to granzyme B+ particles outside of γδD4 cells. Scale bars: 5 μm. **k** Quantification of the number of granzyme B+ particles outside of γδD4 T cells, NKD4 cells or no effector cells counted per A498 target cells, *n* = 6 (42 γδ cells), 3 (15 NK cells), 3 (15 A498 cells without effectors) donors respectively. Individual dots represent the number of particles detected in individual target cells and they are colour-coded according to donor. The box limits represent upper and lower quartiles, and whiskers indicate the minimum and maximum values. All applicable plots show means with standard deviations unless specifically stated otherwise. Statistics were calculated using two-tailed ratio paired *t*-test (**a–c, e, g, h**), two-tailed paired *t*-test (**b, d, i**) and one-way ANOVA (**k**). ns: not-significant.

γδD4 T cells have abundant amounts of granzyme B but limited amounts of perforin (Fig. 1g, i), the latter being synthesised as a precursor cleaved by cathepsins in the maturing lytic vesicles[42,43]. To estimate the extent of killing dependent on perforin, we tested how treatment with the cathepsins-inhibitor E64 affected γδD4 T cell cytotoxicity. First, we confirmed that the E64 inhibitor blocked the

maturation of perforin as the antibody clone δG9, which only recognises the mature form of perforin, showed a decreased signal while granzyme B, FasL and TRAIL were not perturbed (Supplementary Fig. 5b). On-chip killing assay against HMBPP-treated A498GBDR cells showed the same proportion of killers with and without the addition of E64 (Fig. 5b). However, a substantial decrease in granzyme B-mediated

killing was observed with E64, largely compensated by late death receptor-mediated killings (Fig. 5c and Supplementary Fig. 5c, d).

To further test if the availability of perforin was limiting for the cytotoxic potential, sublytic concentrations of recombinant human perforin (rhPrf1) were added during chip killing assays using γδ[D4] T cells against HMBPP-stimulated A498[GBDR]. Our hypothesis was that the sublytic levels of rhPrf1 added at the beginning of the assay would integrate in the membrane of the A498[GBDR] target cells without killing them. Killing of the rhPrf1-preloaded A498[GBDR] target cell would occur only upon γδ T cell degranulation and the delivery of additional perforin and granzyme B as previously reported[44]. The addition of 60 ng/mL rhPrf1 increased overall killing, and this increase was not driven by an increased fraction of killers (Fig. 5d) but instead due to increased serial killing (Fig. 5e, f). No differences were observed in the killing dynamics (Supplementary Fig. 5e, f). Thus altogether, these data suggest that perforin availability could be a bottleneck for serial killing.

### Granzyme B and perforin are differentially regulated in Vγ9Vδ2 T cells

The limited availability of perforin suggests that perforin and granzyme B are differently regulated. Supporting this hypothesis, mRNA levels of perforin and granzyme B were comparable during the γδ T cells expansion while the expression levels of the two proteins differed in the same cells (Fig. 5g, h, Fig. 1g and Supplementary Fig. 5g). This difference was not due to the level of maturation of perforin since antibody clones for both forms showed similar results (Supplementary Fig. 5h). These results suggest that granzyme B and perforin are distinctly regulated after transcription in γδ T cells.

In co-culture assays with HMBPP-treated A498 targets, most γδ[D4] T cells rapidly and continuously degranulated (Supplementary Fig. 5i, j). To further test granzyme B and perforin regulation, protein synthesis was blocked during a 4-h degranulation assay, using cycloheximide. Intracellular levels of granzyme B and perforin were measured at multiple time points. Interestingly, cycloheximide did not influence granzyme B degranulation, showing that granzyme B was not newly synthesised during the assay (Fig. 5i and Supplementary Fig. 5k). About 40% of the total granzyme B was degranulated, and most degranulation occurred within the first 2 h (Fig. 5i, left). Thus, a majority of the granzyme B was conserved intracellularly for γδ[D4] T cells exposed to target cells. In contrast, untreated γδ[D4] T cells replenished the level of perforin during the assay, while cycloheximide-treated cells lost perforin rapidly during the first 2 h of the assay (Fig. 5i, right). This shows that Vγ9Vδ2 T cells use a pre-existing pool of granzyme B that is gradually exhausted upon degranulation but synthesise new perforin to maintain a constant level despite degranulation (Supplementary Fig. 5l). The rate of this synthesis may be insufficient to support prolonged granzyme B-mediated serial killing.

Based on these data we hypothesised that there could be an imbalance between the amount of granzyme B and perforin degranulated in response to target cells. Therefore, γδ[D4] T cells were co-cultured with HMBPP-treated A498 targets for 1.5 h, fixed, and stained for microscopy. Interestingly, granzyme B-positive but perforin-negative particles were found on A498 cells targeted by γδ[D4] T cells. Similar structures were not observed on A498 cells targeted by NK[D4] cells or on A498 target cells cultured alone (Fig. 5j, k). These results suggest that γδ[D4] T cells degranulate granzyme B that cannot penetrate the target cells, possibly due to a lack of perforin in the cytotoxic granules. Alternatively, the excess granzyme B could be dedicated to supramolecular attack particles, a lytic body observed in conventional T cells and NK cells[45,46].

These results show that in response to stimulation γδ[D4] T cells use part of the pre-existing large pool of granzyme B while a fraction of the cells may rely on the formation of new mature perforin.

### Vγ9Vδ2 T cells segregate granzyme B from perforin in lytic vesicles

The differences in regulation and degranulation of granzyme B and perforin highlighted the need to assess how these proteins are distributed in γδ T cell lytic granules. γδ[D4] T cells and NK[D4] cells were fixed and stained for granzyme B, perforin, and CD107a. The latter served as a mask defining the vesicles. As expected from our flow cytometry data, γδ[D4] T cells showed a less balanced phenotype with little perforin, but higher granzyme B content compared to NK[D4] cells (Fig. 6a and Supplementary Fig. 6a). In both conditions, the CD107a+ vesicles showed heterogeneous proportions of granzyme B and perforin signal between individual cells (Supplementary Fig. 6b). The co-occurrence of granzyme B and perforin differed significantly between γδ[D4] T cells and NK[D4] cells (Fig. 6b). In γδ[D4] T cells, perforin was almost always found together with granzyme B (Fig. 6b, Prf-GrzB), while ~65% of granzyme B could be found without perforin (Fig. 6b, GrzB-Prf). In NK[D4] cells, the co-occurrence of granzyme B and perforin was balanced at high levels (Fig. 6b). The previously shown loss of granzyme B expression upon 14-days expansion of γδ T cells was evident from the imaging (Fig. 6c and Supplementary Fig. 6c), and the balance was reversed with co-occurrence of granzyme B with perforin significantly increased (~60%) and perforin no longer exclusively occurring with granzyme B (~40%, Fig. 6d). As donors showed variability in co-occurrence, we wondered if protein amount and co-occurrence could be correlated to γδ[D4] T cells' killing strategy. The granzyme B signal intensity in γδ[D4] T cells did not correlate with the granzyme B-mediated killing capacity in the microwell chip assays, while this correlation was observed for perforin signal intensity (Fig. 6e, f). Furthermore, granzyme B co-occurrence with perforin strongly correlated with the fraction of granzyme B-mediated killing and the number of granzyme B-mediated kills per γδ[D4] T killer cell, with or without HMBPP treatment (Fig. 6g, h). Thus, our analysis shows that the co-occurrence of granzyme B and perforin in the lytic vesicles allows more granzyme B-mediated killing and serial killing in γδ[D4] T cells, while a reduced amount of perforin limits the killing.

## Discussion

Recent studies have highlighted the uniqueness of the γδ T cell plasticity and multifaceted involvement in the body's immune defence[1]. Each individual harbours tissue-specific γδ T cell subsets with γδTCR repertoires that are highly linked to one's past infection history[47–50] and moreover show distinct repertoires infiltrating into tumours[51]. The Vγ9Vδ2 T cell subset which is relatively abundant in peripheral blood, matures, gains potent cytotoxic functions, and expands early after birth[24]. It is likely during this development that a functionally heterogeneous pool of Vγ9Vδ2 T cells is formed and persists into adulthood[32]. Some of these cells might be able to sense normality in somatic cells through weak TCR interactions, as observed in Vδ1 + T cells[52,53] but also recently suggested for the Vγ9Vδ2 TCR[7].

It was previously reported that Vγ9Vδ2 T cells respond to phosphoantigens by proliferating and upregulating granzyme B[15,24]. Studies suggest that Vγ9Vδ2 T cell's innate-like rapid responses might be useful early during various infections by clearing pathogens such as *Plasmodium falciparum*[54], or *Mycobacterium tuberculosis*[36]. Here we show that Vγ9Vδ2 T cells ex vivo rapidly upregulate high levels of granzyme B, and later proliferate upon phosphoantigen and IL-2 stimulation in presence of monocytes. In vivo, monocytes are one of the first responders during infection and may therefore be exposed to phosphoantigens in the nascent inflammation site. Thus, it is possible that phosphoantigen-loaded monocytes activate Vγ9Vδ2 T cells that respond by upregulating granzyme B, which is used in direct and rapid response to pathogens. However, it is unclear if the upregulation of granzyme B and proliferation that we observe in vitro arises with a similar strength and timing in vivo, and if such a dynamic response can

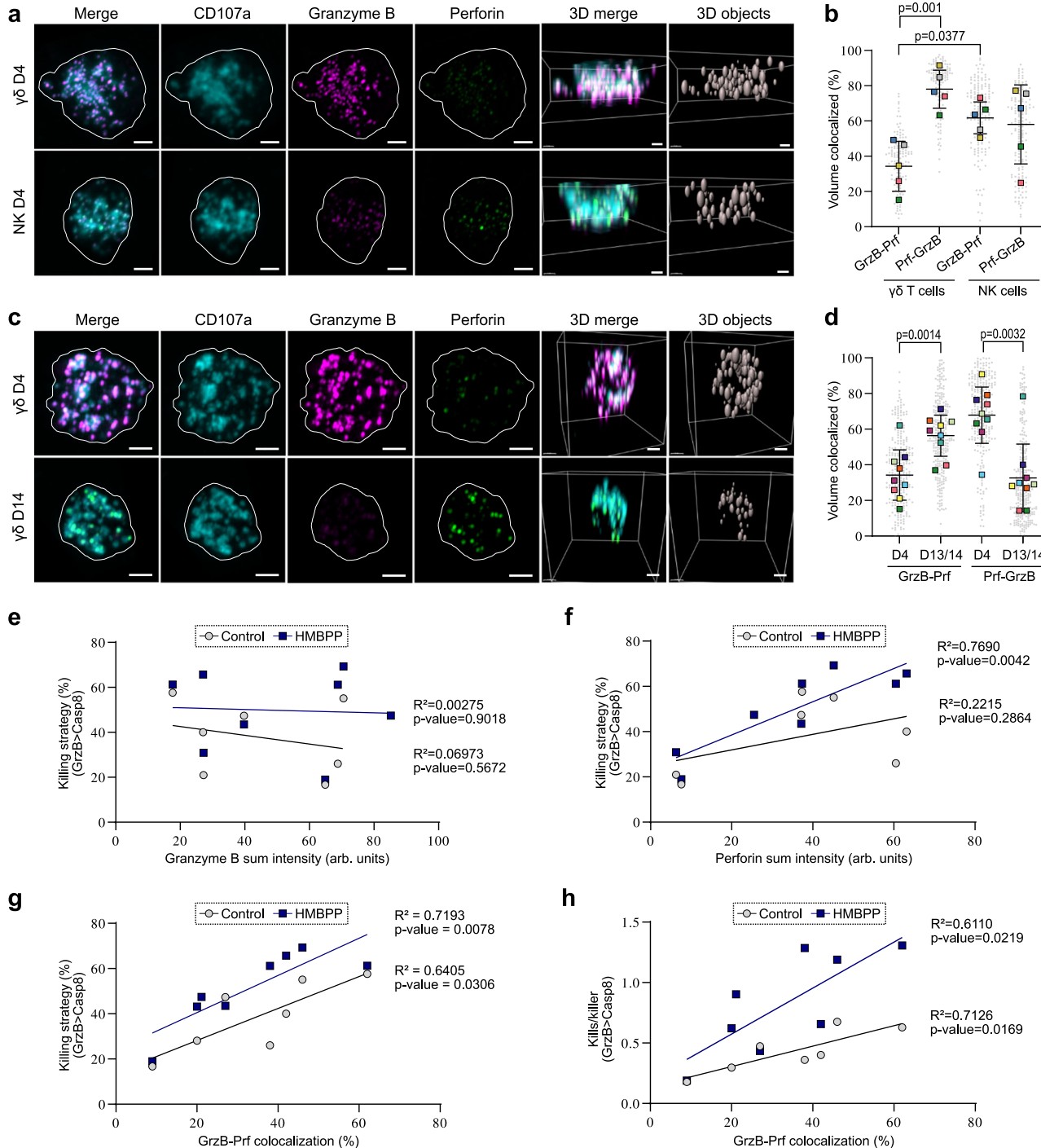

**Fig. 6 | Vγ9Vδ2 T cells segregate granzyme B from perforin in different lytic vesicles. a** Representative Airyscan microscopy images of γδD4 or NKD4 cells from the same donors prepared by immunofluorescence labelling against granzyme B, perforin and CD107a. Scale bars: 2 μm. **b** Proportion of colocalized voxels of granzyme B with perforin (GrzB-Prf) or perforin with granzyme B (Prf-GrzB) in CD107a+ vesicles of γδD4 T cells or NKD4 cells. Square boxes are the means for each donor (n = 5 donors, each ≥15 cells). **c** Representative Airyscan microscopy images of γδD4 or γδD14 T cells from the same donors prepared by immuno-fluorescence labelling against granzyme B, perforin and CD107a. Scale bars: 2 μm. **d** Proportion of colocalized voxels of granzyme B with perforin (GrzB-Prf) or per-forin with granzyme B (Prf-GrzB) in CD107a+ vesicles of γδD4 or γδD13/14 T cells. Square boxes are the means for each donor (n = 9 donors, each ≥15 cells).

**e, f** Correlation between the sum intensity of granzyme B (**e**) or perforin (**f**) in the CD107a+ vesicles with the on-chip percentage of granzyme B-mediated killing of γδD4 T cells (upon HMBPP or not) from the same donors (n = 7). **g** Correlation between the proportion of colocalized voxels of granzyme B with perforin (**b** and **d**) with the on-chip percentage of granzyme B-mediated killing of γδD4 T cells (upon HMBPP or not) from the same donors (n = 7). **h** Correlation between the proportion of colocalized voxels of granzyme B with perforin (**b** and **d**) with the on-chip granzyme B-mediated kills per γδD4 T killer (upon HMBPP or not) from the same donors (n = 7). The lines in **e**–**h** are the linear fittings (with HMBPP treatment or not). All are means with standard deviations. Statistics were calculated using two-tailed paired t-test (**b** and **d**) and simple linear regression (**e**–**h**). ns: not-significant.

also occur in their surveillance against tumours since the latter develop within a different timeframe and microenvironment.

It is also unclear how the phenotypically heterogeneous pool of Vγ9Vδ2 T cells return to steady state after activation. The inability to re-stimulate similar granzyme B upregulation in γδ$^{D14}$ T cells indicates that the initial response to phosphoantigens is unique and irreversible. Alternatively, a resting period longer than 14 days is necessary for the cells to recover the potential to upregulate granzyme B, a possibility that finds support in several studies that found increased cytotoxic functions in Vγ9Vδ2 T cells expanded for longer than 14 days[55]. More efforts are required to fully understand the steady states of the different Vγ9Vδ2 T cell populations and the extent and dynamics of their activation.

Our time-lapse microscopy data show that a significant fraction of the Vγ9Vδ2 T cells are cytotoxic, especially when engaging via the TCR, using a combination of cytotoxic strategies with different dynamics. A large fraction of the cytotoxic Vγ9Vδ2 T cells form long stable contacts with HMBPP-treated targets, sometimes continuing even after target cell death. These contacts seem to be a double-edged sword. On the one hand, HMBPP-treated targets are killed more effectively, and when granzyme B-mediated killing is inhibited by E64, other killing mechanisms could compensate. This is probably facilitated by TCR activation inducing strong and maintained adhesion to target cells allowing time for killing. On the other hand, extended contacts that sometimes last even after target death reduce the mobility and thereby, the possibility for serial killing. In addition, excessive stimulation via TCR has been linked to exhaustion in conventional T cells[56].

Another plausible factor that limits serial killing is the restricted availability of perforin. Donors that show a higher colocalization between granzyme B and perforin kill more effectively with increased serial killing. Granzyme B-mediated killing is also more frequent among serial killers. Consistently, addition of rhPrf1 to the killing assay increased γδ T cell serial killing. While low amounts of rhPrf1 are not sufficient to kill the target cells alone, the added rhPrf1 might remain at the plasma membrane or/and in the endosomal pathway of the target cells. There, rhPrf1 could later facilitate γδ T cell-mediated killing via granzyme B even if the degranulated amount of perforin was low[57]. During co-culture with target cells, γδ T cells are translating new perforin. Thus, it is reasonable to assume that γδ T cells that kill several targets have a bigger pre-existing pool of functional perforin or possibly more effective translation of new perforin. This might be connected to distinct phenotypes or metabolism[58]. The translation of new proteins can be high in energy consumption[59] and therefore serial killing might require additional energy. It has been shown that different γδ T cell subsets have distinct metabolic signatures correlating to their function[60]. However, more investigation is needed to address what makes γδ T serial killers unique and if this potential can be harnessed in vivo, e.g., by increasing γδ T cell's expression of perforin. The unbalanced expression of granzyme B and perforin remains intriguing. It could be a mechanism for the Vγ9Vδ2 T cells to protect themselves and the adjacent healthy tissues from unwanted damage occurring from untargeted degranulation or dying cells. The considerable amounts of granzyme B that do not seem to be immediately allocated to cytolytic functions also suggest that granzyme B might be used in other mechanisms. Granzyme B alone has been shown to play different roles such as extracellular matrix remodelling[61], facilitation of immune cell transmigration[62] and, receptor and cytokines maturation (e.g., cleavage of the IL18 precursor[63]) with relevance in multiple diseases[64]. For instance, extracellular granzyme B degranulated by Vγ9Vδ2 T cells could help shedding of killed cells, as it has recently been observed in vivo for murine intestinal intraepithelial γδ T lymphocytes[65], or facilitate infiltration of other lymphocytes towards the inflammation site by cleaving the extracellular matrix. Our results also suggest that γδ T cells would still degranulate even though vesicles were lacking perforin, as granzyme B-positive but perforin-negative particles were found during co-cultures at the surface of target cells, either in the extracellular space or freshly endocytosed. Although these structures resembled supra-molecular attack particles seen in conventional T cells[45] and NK cells[46], their exact nature and purpose remain to be characterised.

Overall, our study sheds new light on the dynamics of Vγ9Vδ2 T cell cytotoxicity against tumours. After activation, γδ T cells regulate granzyme B and perforin differently in space and time, possibly because the excess of granzyme B has perforin-independent functions or to restrain collateral damage through undesirable cytotoxicity during the early activation phase. Nevertheless, also at early stages of the activation, target cells loaded with phosphoantigens induced the formation of committed effector-target contacts through Vγ9Vδ2-TCR activation, which led to rapid granzyme B-mediated killing when supported by sufficient co-loading of perforin in the lytic vesicles. Vγ9Vδ2 T cells with high perforin availability could go on and kill multiple target cells in sequence. These results show that γδ T cells have a previously unknown cytotoxic potential that may have evolved to fight infection but could be unleashed with improved expansion protocols and harnessed to design more efficient cellular immunotherapies.

## Methods

### Cell lines and culture

Wild-type: A498 kidney carcinoma, K562 lymphoblasts, DLD-1 colorectal adenocarcinoma, Kasumi-1 myeloblast, Daudi B lymphoblasts and HeLa adenocarcinoma were all from ATCC. Wild-type U251 MG glioblastoma was from Sigma Aldrich. A498, K562, DLD-1, Kasumi and Daudi cells were cultured in Gibco RPMI 1640 medium (Fisher Scientific). HeLa cells were cultured in MEME (Sigma Aldrich). U251 were cultured in Gibco DMEM, low glucose, GlutaMAX™ (Thermo Fisher Scientific). All media were supplemented with 10% heat-inactivated FBS (Sigma Aldrich), 1% penicillin-streptomycin (Thermo Fisher Scientific), 1% non-essential amino acid (Sigma Aldrich), and 1% L-glutamine (Sigma Aldrich). Daudi cells were cultured with 10% additional FBS after thawing. The cells were cultured in an incubator at 37 °C and supplemented with 5% $CO_2$. The cultures were regularly checked, and the cells passaged before reaching confluence (2-3 times per week). The absence of mycoplasma in the cell cultures was regularly confirmed using Eurofins Genomics mycoplasmacheck tube service according to the manufacturer's instructions. Our research complies with all relevant ethical regulations. The lab infrastructures for working with BSL2 material were approved by the local work environment authority (Arbetsmiljö verket).

### Generation of stable A498$^{GBDR}$

Wild-type A498 cells (ATCC) were transduced using lentivirus particles to stably express the following gene of the fluorescent reporter: pJML1-NES-LQDT-mGFP-T2A-NES-VGPD-mCherry and sorted by flow cytometry as previously described here[40]. A498$^{GBDR}$ maintenance cultures were regularly selected, adding 2 μg/mL of puromycin (Thermo Fisher Scientific) in the complete RPMI medium 24 h after passaging.

### γδ T cells isolation and characterisation

Anonymous buffy coats from healthy donors were obtained from the department of Klinisk immunologi och transfusionsmedicin in Karolinska Hospital, Huddinge and immediately processed upon arrival. According to current local regulations, working with blood from anonymous healthy human donors requires no ethical permit. Peripheral blood mononuclear cells (PBMC) were harvested from the buffy coat using a Ficoll gradient. Where stated, γδ T cells were immediately extracted from the PBMC by negative selection using TCR γ/δ T Cell Isolation Kit, human (Miltenyi Biotec) according to the manufacturer's instructions. Thawed or freshly prepared PBMCs were cultured at 2×10$^6$ cells/mL in their expansion medium: AIM-V medium

(Thermo Fisher Scientific) supplemented with 10 % heat-inactivated human serum from human male AB plasma (Sigma-Aldrich), 2.5 μM zoledronate (Sigma Aldrich) and 200 IU/mL of recombinant human IL-2 (PeproTech). Expanding cells were re-stimulated with 100 U/mL rhIL-2 every 2 to 3 days. When stated, 10 ng/mL of recombinant IL-15 (R&D System) was used in addition to rhIL-2. 10 nM of (E)−1-hydroxy-2-methyl-2-butenyl 4-pyrop (HMBPP, Sigma Aldrich) was used instead of zoledronate to expand γδ T cells in PBMC cultures, as specifically written in some experiments. On each experiment's day, γδ T cells were freshly extracted from the zoledronate+IL2 treated PBMC culture by negative selection using TCRγδ T Cell Isolation Kit, human (Miltenyi Biotec) according to the manufacturer's instructions. The purification rate of the isolated γδ T cells was assessed every time by flow cytometry, and only the isolations with more than 90% γδ T cells purity and low NK cell contaminants were used in the subsequent assays. Autologous monocytes (used in the minimal expansion of Supplementary Fig. 1j) were isolated using a pan-monocyte isolation kit (Miltenyi Biotec) and seeded on petri dishes with isolated γδ T cells and stimulated with zoledronate and IL2 as described for PBMCs above.

### Immunostaining for flow cytometry

For phenotyping, PBMCs or expanded/isolated γδ T cells were stained for 10 min at RT in PBS with antibodies at the listed concentration and a fixable viability dye at 1:1000 (510 or AF700 from BD Bioscience) and washed. For intracellular staining, the cells were fixed with a fix/perm kit according to the manufacturer´s instructions (BD Biosciences) and stained with antibodies diluted at the due concentration in the kit's buffer for 10 min at RT. Compensation of the full panel was performed for each experiment using the cells or CompBeads (BD Biosciences) and an unstained sample was included as a control. Flow cytometry data was acquired on the FACS Canto II system (BD Biosciences), and analysis was performed using FlowJo 10.7.1 (FlowJo LLC). The Vγ9Vδ2 T cells size fluctuation during the expansion was compensated by dividing fold-change values by their corresponding FSC-A value at each time point. This enabled us to compare the signal/molecular densities per cell rather than the total amount of signal/molecules. All antibodies used in this study are described (name, conjugate, provider, dilution) in the supplementary table 1. Dimensional reduction (10 donors): each dataset was down-sampled using the plugin DownSampleV3 (to obtain an equal number of measurements) prior to concatenation and processing using the plugin UMAP. Clustering and subpopulation visualisation was performed using the FlowSOM plugin on the described markers.

### CD107a degranulation assays

A498 wild-type cells were counted and seeded for 24 h in a flat bottom 96-well plate (Sigma Aldrich) in complete RPMI medium. Isolated Vγ9Vδ2 T cells (from 4-days or 14-days expanded cultures) were counted and re-suspended in fresh complete AIM-V medium. For the assay, the A498 medium was replaced by complete AIM-V medium and when stated HMBPP (1.5 μM) was added to the cells. The Vγ9Vδ2 T cells were then seeded on the target cells at a 1:1 E:T ratio and co-cultured for 4 h at 37 °C and 5% CO2 in the incubator. After 1 h, 1 μL of anti-CD107a (conjugated to BV421 or AF647, BD Biosciences) was added to each well. When stated, monensin (Golgi STOP, BD Biosciences) was added after 1 h to the wells at the manufacturer's concentration. When stated cycloheximide (Sigma Aldrich) was added at 2 μg/mL to the corresponding wells at the start of the assay. For degranulation experiments with multiple time points, a reverse time course was implemented (differed starts at 4 h, 3 h, 2 h, …) to enable to stain all samples at the same end point. At the end of the assay, all cells were detached using PBS-EDTA for 15 min at 37 °C and transferred to a v-bottom 96-well plate (Sigma Aldrich) for staining. All wells were washed with PBS, stain for the surface markers (including the TCRVδ2) and with a fixable viability dye, prior to fixation,

intracellular staining and measurement by flow cytometry as described above.

### FACS killing assay

Daudi cells were first stained using CellTrace FarRed (Invitrogen) following the manufacturer's protocol, washed, counted, and re-suspended in complete AIM-V medium for the assay. Isolated Vγ9Vδ2 T cells (from 4-days or 14-days expanded cultures) were counted and re-suspended in fresh complete AIM-V medium. Effectors and targets were mixed at the mentioned E:T ratio in flow cytometry sterile tubes and co-cultured for 4 h in the incubator. Single cultures were performed to assess spontaneous death. At the end of the assay, the samples were washed and stained for the surface markers (incl. TCRVδ2) and a viability dye. The samples were kept on ice and immediately measured by flow cytometry. The killing rate was established as the percentage of dead Daudi cells minus the percentage of their spontaneous death.

### Time-lapse microscopy in the microwell chip

A498GBDR target cells were seeded into the microwell chip in complete RPMI medium 24 h before the killing assay. Before seeding γδ T cells, the medium in the microwells was changed to complete AIM-V medium. The positions to image in the microwell chip platform were set and scanned first to assess the initial spontaneous death of the targets. Then the freshly isolated γδ T cells were counted and added into the chamber resulting in stochastic distribution. HMBPP was added at 1.5 μM when indicated. Images were immediately acquired after seeding the γδ T cells with a 10x Plan-Apochromat objective on a Zeiss Colibri 7 wide-field microscope (Zeiss, Germany) with an incubation chamber at 37 °C and 5 % CO2 every 3 min for 16 h covering >2000 hexagonal microwells/condition.

**TCR blocking.** The isolated Vγ9Vδ2 T cells were pre-blocked for 30 min with 5 μg/mL of anti-TCR (clone IMMU510, Beckman Coulter). Then at the start of the killing assay, the anti-TCR was added at 5 μg/mL in the medium of the corresponding condition.

**Addition of recombinant perforin.** The killing assay with isolated Vγ9Vδ2 T cells and A498GBDR was performed by adding of human recombinant perforin (hrPrf1, Abbexa). 60 ng/mL of rhPrf1 appeared to be the sublytic threshold of rhPrf1 as separately determined using serial dilutions of rhPrf1 added in the medium of A498GBDR. This was in agreement with previously reported literature values (30-120 ng/mL[66]). Upon higher lethal doses of hrPrf1, A498GBDR cells were rapidly lysed without showing a nuclear signal of the dual reporter construct. γδD4 T cells were not affected by the addition of 60 ng/mL rhPrf1.

**Inhibition of perforin.** The isolated expanded Vγ9Vδ2 T cells were pre-incubated for 24 h with the E64 inhibitor (Sigma Aldrich) at 30 μM (final concentration) in the complete AIM-V medium. Before the killing assay, a fraction of the Vγ9Vδ2 T cells was retrieved and prepared as described above for flow cytometry measurement of the indicated protein of interest expression levels. The remaining Vγ9Vδ2 T cells were seeded on A498GBDR in the microwell chip platform and E64 was added in the medium of the corresponding conditions at 30 μM (final concentration) at the start of the assay.

### Image analysis of the killing assay in microwell chip

To determine the type of death of A498GBDR target cells, the imaged time-lapse was analysed manually frame by frame. The brightness of each channel was adjusted to obtain uniform signal in the cytosol of each A498GBDR cell. Each target cell was followed over the full experiment time, and frames for contact initiation, reporter cleavage, time of death, and γδ T cells disassociation from the target were reported. For serial killing analysis, γδ T cells were observed from the time of

seeding, and contacts with target cells, commitment, and induction of death or not were described.

Representative examples of the events observed in the killing assay in the microwell platform were exported as supplementary movies and annotated using ImageJ and OpenShot software.

## Airyscan microscopy

Glass coverslips (18 mm square, #1.5, Marienfield) were coated with Corning™ Cell-Tak Cell and Tissue Adhesive (Fisher Scientific) at a concentration of 4.5 μg/cm² according to the manufacturer's protocol. Isolated γδ T cells were seeded on coated coverslips to adhere for 20 min at 37 °C. Cells were fixed and permeabilized for 10 min with BD Cytofix/Cytoperm™ (BD Biosciences) and fluorescently labelled with primary and secondary antibodies against perforin (clone δG9, BioLegend), granzyme B (clone GB11, BD Biosciences) and CD107a (clone H4A3, BD Biosciences). Coverslips were mounted with ProLong™ Glass Antifade (Thermo Fisher Scientific) and imaged the following day. Images were acquired using a confocal microscope LSM880 (Zeiss) with a 63x/1.4 Plan-Apochromat oil-immersion objective and Airyscan detector. The images were then processed using the in-built function (Airyscan 3D mode) of the ZEN 2.3 SP1 software (Zeiss). For technical reasons, γδ T cells were expanded for either 13 or 14 days prior to the Airyscan imaging of lytic vesicles (Fig. 6c, d) and pooled together for analysis (referred as γδ^D13/14).

For detection of the extracellular particles, A498 wild-type cells were used as targets and seeded on glass coverslips for 24 h. Isolated 4-days expanded Vγ9Vδ2 T cells were co-cultured for 1.5 h with the targets in the presence of 1.5 μM of HMBPP. Treated-A498 targets without Vγ9Vδ2 T cells were used as the negative control. After the incubation, the cells were fixed and stained as described above. Perforin and granzyme B were used at the previously mentioned concentration. Phalloidin-AF568 (Thermo Fisher Scientific) was used at 1 unit/sample according to the manufacturer's protocol. The Airyscan microscopy was performed as mentioned above and the granzyme B-positive extracellular particles were counted manually on each A498 target cell using the same image settings in all conditions.

## Image analysis of the lytic vesicles

The images acquired with Airyscan microscopy were processed with Imaris 9.5 (Oxford Instruments) and ImageJ. The background was removed using Imaris background subtraction method (sigma = 0.25), a baseline was subtracted, and the channel shifts were adjusted. For colocalization analysis between granzyme B and perforin, CD107a channel was used as a mask and thresholding was performed manually (one threshold for all conditions in each experimental dataset). For the analysis of objects within the cell, channels for granzyme B and perforin were added using ImageJ and subsequently objects were defined using the spots method in Imaris. Thresholding was performed manually as stated above. Fluorescence sum intensities were normalised to the standard deviation of all pooled objects in each dataset (e.g., γδ^D4 and γδ^D14) to account for staining differences between experiments.

## Electron microscopy

Vγ9Vδ2 T and NK cell pellets were fixed in 0.1% glutaraldehyde and 4% paraformaldehyde in 0.1 M PHEM buffer (60 mM PIPES, 25 mM HEPES, 10 mM EGTA, and 2 mM MgCl₂ at pH 6.9) overnight, and subsequently embedded in 10% gelatin, infiltrated with 2.3 M sucrose and frozen in liquid nitrogen (LN₂). Samples were stored in LN₂ until sectioning. Ultrathin sections of 70-80 nm were cut with a Leica Ultracut (equipped with UFC cryochamber) at −110 °C and picked up with 50:50 mixture of 2.3 M sucrose and 2% methyl cellulose.

Sections were labelled with rat anti-human Granzyme B (clone 496B, eBioscience) or mouse anti-human CD63 (DSHB, USA) or mouse anti-human Chondroitin Sulfate 4 (AMSBIO), followed by a bridging rabbit-anti-rat antibody (Jackson Immuno Research Laboratories Inc.) or rabbit-anti-mouse antibody (Rockland Immunochemicals) and 10 nm protein A gold (University Medical Centre, Utrecht, The Netherland). Samples for Chondroitin Sulfate 4 staining were pretreated with chondroitinase ABC (AMS.E1028-02, AMSBIO) for 2 h at 37 °C according to the manufacturer's instructions. For each sample, cells were chosen for imaging by using systematic uniform random sampling[67]; a starting point was assigned (e.g., the corner of the EM-grid) on a section and every fourth cell was imaged (the cells orientation is random as the EM-block comes from a cell pellet). 20 cells were imaged per donor for each condition. Microscopy was done using JEOL JEM1230 at 80 kV and images acquired with a Morada camera. The dense-core secretory granules were recognised by morphology. Quantification was done using Image J.

## PrimeFlow and qPCR

The mRNA level measurements using the PrimeFlow method were conducted by precisely following the manufacturer's detailed protocol (PrimeFlow RNA assay kit, Thermo Fisher Scientific). The PrimeFlow probe sets were: PRF1-AF750 (VA6-13185, Thermo Fisher Scientific) and GZMB-AF488 (VA4-3084452, Thermo Fisher Scientific) and the antibodies: Anti-Perforin-Pacific-Blue; anti-granzyme B-AF647; anti-Vδ2TCR-PE and anti-Vγ9TCR-PE-Cy7 and the fixable viability BV510.

For qPCR measurements: the mRNA was extracted from isolated NK cells or isolated γδ T cells using the RNEasy kit (QIAGEN) and cDNA was made using the iScript cDNA synthesis kit (Bio-Rad laboratories). Quantitative PCR was performed using the SSoAdvanced Universal SYBR Green Supermix (Bio-Rad laboratories) with the following primers: granzyme B: 5′- CGACAGTACCATTGAG TTGTGCG-3′ and perforin: 5- ACTCACAGGCAGCCAACTTTGC-3′. Samples were run using the CFX384 Touch Real-Time PCR Detection System (BioRad Laboratories). β−2-microglobulin (β2M) mRNA was used as a housekeeping gene to normalise the mRNA levels of granzyme B and perforin.

## Re-stimulation of expanding γδ T cells

Whole PBMCs from healthy donors were first stained using CellTrace Violet (CTV, Invitrogen) according to the manufacturer's protocol. An unstained vial of the PBMC was frozen (in human serum and 10% DMSO) for later. The cells were stimulated with zoledronate and IL-2 to expand Vγ9Vδ2 T cells (protocol above), and aliquots of the culture were frozen at the indicated time points (0 to 14 days). After 14-days, the expanded cells were stained again with CTV using the same procedure. The initially frozen PBMC vial was thawed and mixed with the 14-days expanded CTV + Vγ9Vδ2 T cells (~5% of the new culture) and re-stimulated with the same concentrations of zoledronate and IL-2 for 14-days. Aliquots were frozen at the indicated time points (14 to 28 days). At the end of the second expansion, all samples were thawed and stained (surface and intracellular) simultaneously and measured using flow cytometry.

## Statistical analysis

Statistical analyses were performed in GraphPad (Prism 9) using a two-tailed (paired when appropriate) Student's t-test, Kruskal−Wallis test, or one-way ANOVA as indicated. Data are represented as means ± standard deviations or median as indicated. ns: not significant.

## Reporting summary

Further information on research design is available in the Nature Portfolio Reporting Summary linked to this article.

## Data availability

All data in this study are available within the article and its supplementary files and from the corresponding author upon reasonable request. Source data are provided with this paper.

## Code availability

No homemade code was used in this study.

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

## Acknowledgements

We thank The Knut and Alice Wallenberg foundation (Grant No 2018.0106), The Swedish research council (Grant No 2019-04925), The Swedish foundation for strategic research (Grant No SBE13-0092), The Swedish childhood cancer fund (Grant No MT2019-0022), The Swedish cancer society (Grant No 19 0540 Pj) for financial support. This work was partially supported by the Research Council of Norway through its Centres of Excellence scheme, project number 332727. We also thank Juan Basile and the flow cytometry facility of the Karolinska Institutet, and Albert Blanchart Aguado and the VirusTech Core Facility at Karolinska Institutet.

## Author contributions

P.A.S., S.T., K.K., E.A., M.U., B.Ö. conceptualised the study. P.A.S., K.K., E.K.S., E.E. and A.K.W. performed experiments with the help of S.T., Q.V. and A.B. K.K. performed the Airyscan microscopy. N.S. designed and prepared microwell chips. E.K.S. performed the electron microscopy with input from A.B. and K.J.M. E.E. performed the qPCR. C.W. developed the fluorescent reporter and provided important input. P.A.S., K.K. and B.Ö. wrote the manuscript with input from all the authors. P.A.S. and B.Ö. supervised the overall study.

## Funding

## Competing interests

The authors declare no competing interests.
