## [Peer Review File · Nature Communications]

Modulation of lytic molecules restrain serial killing in $\gamma\delta$ T lymphocytesEditorial Note: Parts of this Peer Review File have been redacted as indicated to maintain the confidentiality of unpublished data.

REVIEWER COMMENTS

Reviewer #1 (Remarks to the Author):

In this study, the authors describe the regulation of cytotoxic molecules in human Vg9Vd2 T cells that have been expanded *ex vivo* from PBMCs. They show that granzyme B in particular increases after 3 days into expansion, but then decreases by day 14, while perforin expression remains the same, and granulysin increases over 2 weeks of expansion. Vg9Vd2 T cells used granzyme B-mediated killing when expanded 3-4 days, whereas this was less pronounced with cells expanded for 2 weeks. HMBPP-treated Vg9Vd2 T cells were better at killing target cells than untreated control Vg9Vd2 T cells, and serial killing occurred more often in HMBPP-treated Vg9Vd2 T cells. This was TCR dependent. The lack of serial killing was due to deficient perforin pools – restricted availability of perforin -- because addition of recombinant perforin increased serial killing. The authors found that granzyme B was stored in Vg9Vd2 T cells, but protein translation of perforin was necessary for serial killing. Finally, co-occurrence of granzyme B and perforin in lytic vesicles endowed Vg9Vd2 T cells with serial killing ability.

This is an excellent study which combines immunology and cell biology. It contains unique insights into the cytotoxic behavior of Vg9Vd2 T cells that have broad implications for cancer immunotherapy. To my knowledge, no study has gone this much in depth for mechanistic killing by Vg9Vd2 T cells. The mechanisms of degranulation and synthesis of lytic molecules by Vg9Vd2 T cells shown herein stand in clear contrast to Gillian Griffiths work on serial killing by CD8 T cells, which are much better serial killers than Vg9Vd2 T cells. Thus, the data provide mechanistic answers for the differences between gdT cells and CD8 T cells. At the same time, differences with NK cells are also provided. The experiments are well controlled with loss-of-function and gain-of-function approaches across multiple human donors. The statistics are appropriate. The use of advanced imaging techniques both at cellular and molecular levels is a particular highlight of the study. It will be interesting as the authors suggest to determine whether the restricted perforin availability is inherent to specific subsets of Vg9Vd2 T cells or whether this is an acquired trait.

My only suggestion to improve the manuscript is to state more specifically how the data are displayed in Figure 2B,C,E where it says 3 donors were used but 100s of dots are shown. A similar request can be made for Figure 3H, Figure 5K.

Reviewer #2 (Remarks to the Author):

In “Modulation of lytic molecules restrain serial killing in $\gamma\delta$ T lymphocytes”, the authors provide an in-depth dissection of the kinetics and mechanisms of killing of tumor targets by $\gamma\delta$ T cells. While $\gamma\delta$ T killing has not been as well described as NK cell or CD8+ T cell killing, here the authors show that activation in culture leads to the expansion of a population of cells at day 4 that relies on granzyme B-mediated cytotoxicity but whose killing capacity is apparently limited by the production of perforin. Overall, the paper is excellent and sheds light on a previously poorly studied population of immune effector cells. Some areas that could use additional clarification or support are described below.

Major comments:

1. It is unclear what the population level dynamics are that are being shown particularly in Fig 1. The UMAP in Fig 1d is not very informative without showing the respective expression of each of the markers that have contributed to it. What is the origin of the more heterogeneous population at day 14? Do the cells at day 3 continue to differentiate as they expand on days 3-14 or are there residual cells that expand at later time points?
2. How was the number of granules per cell enumerated? Extended data 2a suggests ~5 granules per cell but the microscopy data in Fig 6 suggests far more than that. While I appreciate these are different imaging techniques so may not directly correlate, this should be clarified. Close reading of the text suggests that the EM granule enumeration was per transversal section of the cell, in this case the average number of granules per cell measurement maybe can't be inferred by a single slice.

3. It is also unclear how cells were selected for EM analysis. Based on Fig 1 the cells at day 4 were likely phenotypically all similar, but how many cells were imaged? Given the heterogeneity of granzyme content and the heterogeneity of cells at day 14, better clarity on the source of cells selected for the very low through-put approaches (EM) would be helpful and it should be stated how many cells were imaged by EM.

4. The finding that the addition of soluble recombinant perforin increases the capacity for serial killing is surprising (fig 5e) as it is unclear to me how the authors propose this is functioning in the context of an immunological synapse. Where and how is it expected that granzyme and perforin would be available for the T cell to utilize these together to increase serial killing capacity? Given the other data presented, it seems clear that availability of perforin is limiting serial killing capacity, but overexpression of perforin within the $\gamma\delta$ T cell would be a more intuitive way to compensate for limited perforin availability as opposed to adding soluble perforin. Further, is the "other" form of killing also increased in the presence of soluble perforin, and if so, what does this suggest about this mechanism?

Minor:

1. Fig 1a – if days 3-14 are not statistically significant, is there significance between day 0 and later time points? If so, consider marking this.

2. Abstract line 29 – add "by $\gamma\delta$ T cells" after "serial killing of tumor cells" to clarify?

Reviewer #3 (Remarks to the Author):

The authors present a detailed, multifaceted characterization of $\gamma\delta$ T cell activation. These cells respond to signals in a manner complementary to $\alpha\beta$ T cells, and provide both a complementary physiology that is important to the immune response and potential tools for cellular immunotherapy.

This report applies a range of biomarker, cell expansion, and cell function assays to understand $\gamma\delta$ T cells following activation, predominantly zoledronate stimulation of PBMCs. Overall, the methods are sound and the selection of tools appropriate for the question being asked. However, selection of some outputs is puzzling. In particular, the use of a contour plot to present proteomic data in a UMAP analysis (where data spacing is adjusted, Fig. 1d), is odd and impairs analysis. Similarly, the peaks in Fig. 1b are not evenly spaced, particularly for low levels of fluorescence.

As stated as a goal for the study, this report provides a cohesive framework for understanding functional responses of T cells. This report does provide new insight into changes in cell response over the course of activation and population expansion. However, this depth comes with a tradeoff in understanding how multiple types of cells affect $\gamma\delta$ T cell response. It is in fact surprising that the data of cell response is rather reproducible across multiple donors. The $\beta\gamma$ T cells represent a small fraction of PBMC, so the impact of additional cell types during activation should be discussed.

Point-by-point reply

First of all, we would like to express our gratefulness to all three reviewers for providing positive and constructive feedback. We believe that by addressing the comments we have significantly improved the manuscript. We have made clarifications to the text and figures, and added some new results as described in the point-by-point reply below:

Reviewer #1 (Remarks to the Author):

In this study, the authors describe the regulation of cytotoxic molecules in human Vg9Vd2 T cells that have been expanded ex vivo from PBMCs. They show that granzyme B in particular increases after 3 days into expansion, but then decreases by day 14, while perforin expression remains the same, and granzyme B increases over 2 weeks of expansion. Vg9Vd2 T cells used granzyme B-mediated killing when expanded 3-4 days, whereas this was less pronounced with cells expanded for 2 weeks. HMBPP-treated Vg9Vd2 T cells were better at killing target cells than untreated control Vg9Vd2 T cells, and serial killing occurred more often in HMBPP-treated Vg9Vd2 T cells. This was TCR dependent. The lack of serial killing was due to deficient perforin pools – restricted availability of perforin -- because addition of recombinant perforin increased serial killing. The authors found that granzyme B was stored in Vg9Vd2 T cells, but protein translation of perforin was necessary for serial killing. Finally, co-occurrence of granzyme B and perforin in lytic vesicles endowed Vg9Vd2 T cells with serial killing ability.

This is an excellent study which combines immunology and cell biology. It contains unique insights into the cytotoxic behavior of Vg9Vd2 T cells that have broad implications for cancer immunotherapy. To my knowledge, no study has gone this much in depth for mechanistic killing by Vg9Vd2 T cells. The mechanisms of degranulation and synthesis of lytic molecules by Vg9Vd2 T cells shown herein stand in clear contrast to Gillian Griffiths work on serial killing by CD8 T cells, which are much better serial killers than Vg9Vd2 T cells. Thus, the data provide mechanistic answers for the differences between gdT cells and CD8 T cells. At the same time, differences with NK cells are also provided. The experiments are well controlled with loss-of-function and gain-of-function approaches across multiple human donors. The statistics are appropriate. The use of advanced imaging techniques both at cellular and molecular levels is a particular highlight of the study. It will be interesting as the authors suggest to determine whether the restricted perforin availability is inherent to specific subsets of Vg9Vd2 T cells or whether this is an acquired trait.

My only suggestion to improve the manuscript is to state more specifically how the data are displayed in Figure 2B,C,E where it says 3 donors were used but 100s of dots are shown. A similar request can be made for Figure 3H, Figure 5K.

We thank the reviewer for the excellent and complete summary of our work and for the suggestions of how to improve the figures. As suggested for Fig. 2B,C,E, we now display dots representing individual granules, which are colour-coded to represent the different donors. We have also updated the figure captions providing more details about the number of dots and donors, and clarified what is shown in the graphs. Fig. 3H and Fig. 5K have also been updated as suggested.

Reviewer #2 (Remarks to the Author):

In “Modulation of lytic molecules restrain serial killing in $\gamma\delta$ T lymphocytes”, the authors provide an in-depth dissection of the kinetics and mechanisms of killing of tumor targets by $\gamma\delta$ T cells. While $\gamma\delta$ T killing has not been as well described as NK cell or

CD8+ T cell killing, here the authors show that activation in culture leads to the expansion of a population of cells at day 4 that relies on granzyme B-mediated cytotoxicity but whose killing capacity is apparently limited by the production of perforin. Overall, the paper is excellent and sheds light on a previously poorly studied population of immune effector cells. Some areas that could use additional clarification or support are described below.

We are delighted that the reviewer appreciates the work and the findings described in the manuscript. We have clarified the text and the figures according to the reviewer's suggestions and we performed several experiments in order to answer the remaining questions.

Major comments:

1. It is unclear what the population level dynamics are that are being shown particularly in Fig 1. The UMAP in Fig 1d is not very informative without showing the respective expression of each of the markers that have contributed to it. What is the origin of the more heterogeneous population at day 14? Do the cells at day 3 continue to differentiate as they expand on days 3-14 or are there residual cells that expand at later time points?

We indeed provided a simplified UMAP graph as Fig. 1 was already relatively dense and the remaining space limited. Nevertheless, we understand the concern of reviewer #2 and we have now included more information in the UMAP graph. We performed a FlowSOM clustering analysis of the subpopulations driving the dynamics between day 0 to 14 using the mentioned markers and highlighted 6 populations in the new panel f of Fig. 1. A legend explaining the expression pattern in each subpopulation is included. Additionally, we provide the distribution of each marker on the UMAP graphs in a new panel in Extended Data Fig. 1d. We believe that including this information clarifies the observed expression dynamics during the expansion.

2. How was the number of granules per cell enumerated? Extended data 2a suggests ~5 granules per cell but the microscopy data in Fig 6 suggests far more than that. While I appreciate these are different imaging techniques so may not directly correlate, this should be clarified. Close reading of the text suggests that the EM granule enumeration was per transversal section of the cell, in this case the average number of granules per cell measurement maybe can't be inferred by a single slice.

The granules, as presented in Extended Data Fig. 2a, were indeed counted in individual cell sections using 2-dimensional transmission electron microscopy (resolution of a few nanometers) while the data in Fig. 6 is based on 3-dimensional Airyscan fluorescence microscopy (lateral resolution of ~150 nm). Thus, the number of granules in Extended Data Fig. 2a do not represent the total granules content of the cell. We have now clarified this in the text (LINE 185), on the y-axis of the Extended Data Fig. 2a and in the figure captions.

3. It is also unclear how cells were selected for EM analysis. Based on Fig 1 the cells at day 4 were likely phenotypically all similar, but how many cells were imaged? Given the heterogeneity of granzyme content and the heterogeneity of cells at day 14, better clarity on the source of cells selected for the very low through-put approaches (EM) would be helpful and it should be stated how many cells were imaged by EM.

We have now added more details about the number of cells analysed by TEM in the methods section and in the figure captions (20 cells per donor for each condition). We also provide more information in the methods section about how the cells were selected based on systematic uniform random sampling.

4. The finding that the addition of soluble recombinant perforin increases the capacity for serial killing is surprising (fig 5e) as it is unclear to me how the authors propose this

is functioning in the context of an immunological synapse. Where and how is it expected that granzyme and perforin would be available for the T cell to utilize these together to increase serial killing capacity? Given the other data presented, it seems clear that availability of perforin is limiting serial killing capacity, but overexpression of perforin within the $\gamma\delta$ T cell would be a more intuitive way to compensate for limited perforin availability as opposed to adding soluble perforin. Further, is the “other” form of killing also increased in the presence of soluble perforin, and if so, what does this suggest about this mechanism?

These are all very valid questions. In agreement with reviewer #2, we initially thought that overexpression of perforin would be more suitable to compensate for the limited perforin availability. However, transduction of non-expanded primary human V γ 9V δ 2 T cells is challenging and therefore not often performed. Furthermore, the mRNA of perforin is present and upregulated on day 3 similarly as granzyme B (Fig. 5h) which suggests that perforin is modulated at the post-translational level. In such case, overexpression of perforin might not strongly impact the perforin protein level. For this revision, we have developed a transduction protocol using lentivirus for the expression of human perforin in primary human V γ 9V δ 2 T cells on day 4 of their expansion and tested it on six donors as shown in the supplementary data for the reviewers-only; Fig. R1a-e (at the end of this document). Perforin (or the mock control) was co-expressed with a GFP gene downstream of an IRES promoter (expected to give a 1:1 ratio between perforin and GFP at the mRNA level). Although the overall transduction efficiency remained low, we observed that GFP was well expressed on a fraction of primary human V γ 9V δ 2 T cells, a population we could retrieve using FACS sorting. However, these GFP+ V γ 9V δ 2 T cells did not express significantly more perforin than mock-transduced cells (Fig. R1a-c, for reviewers only shown below). Moreover, the GFP+ V γ 9V δ 2 T cells often showed an altered (sometimes reduced) granzyme B expression level (Fig. R1c). FACS-sorted GFP+ V γ 9V δ 2 T cells (perforin and mock) from three donors were used in chip killing assays against A498^{GBDR} (same approach as we used in other experiments shown in the manuscript). Although the GFP+ V γ 9V δ 2 T cells were active and cytotoxic (Fig. R1e), only a trend of enhanced granzyme B-mediated killing was observed (Fig. R1d), most likely because of the limited increase in perforin expression and varied granzyme B amounts mentioned above. Although encouraging, we have decided not to include these data in the manuscript since significant optimization is required to increase transduction efficiency and post-translational expression of the transduced perforin. Our ambition is to pursue this in a separate project.

With our initial approach of adding recombinant human perforin (rhPrf1), our hypothesis was that the sublytic levels of rhPrf1 added at the beginning of the assay would integrate in the membrane of the A498 target cells without killing them. Killing of the rhPrf1-preloaded A498 target cell would occur only upon $\gamma\delta$ T cell degranulation and the delivery of additional perforin and granzyme B. Offering some support to this hypothesis, it has previously been shown that high doses of recombinant perforin can directly kill target cells while lower doses are non-lytic unless combined with recombinant granzyme B (Waterhouse et al., *Journal of Cell Biology*, 2006; doi.org/10.1083/jcb.200510072).

In an attempt to test the hypothesis further and shed light on the mechanism, we performed additional experiments during the revision period. These experiments were designed and evaluated based on two models that have been proposed for perforin/granzyme b mediated killing (Prager & Watzl, *Journal of Leukocyte Biology*, 2019; doi.org/10.1002/JLB.MR0718-269R). Model 1) Perforin degranulated at the immunological synapse at high local concentration form a pore in the plasma membrane of the target cell, which enables the degranulated granzyme B to penetrate the target cell and initiate the apoptosis cascade. Model 2) Both degranulated perforin and granzyme B, are endocytosed by the target cell and accumulate in the early endosomes and multivesicular bodies. There, perforin proteins form the pores that would enable granzyme B to enter the target cell's cytosol and initiate apoptosis.

To our knowledge, most of the studies using recombinant perforin and granzyme B to understand their function have focused on the localization of granzyme B within target cells rather than visualizing perforin (e.g., Thiery et al., Nat. Immunol., 2011; doi.org/10.1038/ni.2050). The aim of our additional experiment was therefore to assess where hrPrf1 would localize after its addition to the target cells. First, we confirmed that higher concentrations of perforin (> 60 ng/mL) induced rapid lysis of the target cells also in the absence of effector cells (Fig. R2a). In the following experiments, sublytic (60 ng/mL) rhPrf1 (or none for the control) was added to A498 target cells (without $\gamma\delta$ T cells) for different time points (1, 5, 15, 30, 60 and 120 min). The cells were fixed, permeabilized and stained for perforin with different antibody clones (δ G9 and B-D48), and for early endosomes (using anti-EEA1). We used two different clones to increase the chances of detection since their respective epitopes, that are not disclosed by the vendors, could be masked once the perforin is integrated in the membrane and the pore is formed (Ivanova et al., Science Advances, 2022: doi.org/10.1126/sciadv.abk3147). Our experiment showed weak staining for perforin on vesicular structures in some of the treated target cells but a more diffuse signal on the target cells that did not receive perforin (Fig. R2b). The perforin staining did not co-localize with the early endosome marker in any of three replicates investigated. As the local density of rhPrf1 added at sublytic levels is expected to be very low on A498 target cells, an additional experiment was conducted adding 1000 ng/mL perforin for only 5 min prior to fixation and the same immunofluorescence staining procedure. This short incubation time did not kill all the cells despite the high concentration of rhPrf1. This resulted in more perforin staining in vesicular structures which still did not co-localize with the early endosomal marker (Fig. R2b). We conclude that if sublytic levels of perforin are taken up by target cells, it is difficult to detect by fluorescence microscopy of immunolabelled perforin. Thus, these new experiments unfortunately did not provide additional understanding of the mechanism. As mentioned earlier, our best hypothesis is that the added rhPrf1 integrates in the membrane(s) of the A498 target cells and supports its killing when a $\gamma\delta$ T cell attacks by degranulating granzyme B. We now clarify this hypothesis in the text of the manuscript and discuss our results of Fig. 5 in the context of the two mechanistic models for perforin-mediated killing (LINES 385-389 and 534-538).

Minor:

1. Fig 1a – if days 3-14 are not statistically significant, is there significance between day 0 and later time points? If so, consider marking this.

Day 0 is indeed statistically different from the later time points. We have added the respective comparisons in Fig. 1a, updated the Fig. 1 caption and included the ANOVA statistical analysis with the significant p-values in the raw data excel file.

2. Abstract line 29 – add “by $\gamma\delta$ T cells” after “serial killing of tumor cells” to clarify?

We have included “by $\gamma\delta$ T cells” at the end of this sentence as suggested.

Reviewer #3 (Remarks to the Author):

The authors present a detailed, multifaceted characterization of $\gamma\delta$ T cell activation. These cells respond to signals in a manner complementary to $\alpha\beta$ T cells, and provide both a complementary physiology that is important to the immune response and potential tools for cellular immunotherapy.

This report applies a range of biomarker, cell expansion, and cell function assays to understand $\gamma\delta$ T cells following activation, predominantly zoledronate stimulation of PBMCs. Overall, the methods are sound and the selection of tools appropriate for the question being asked.

However, selection of some outputs is puzzling. In particular, the use of a contour plot to present proteomic data in a UMAP analysis (where data spacing is adjusted, Fig. 1d), is odd and impairs analysis.

We are pleased that the reviewer found our methods appropriate. Figure 1d shows the UMAP analysis of the expression by $\gamma\delta$ T cells of a restricted panel of common markers including CD27, CD28 and CD16, that were measured by flow cytometry. As this was confusing, we have now clarified it in the text (LINES 111-113), in the caption of Fig. 1 and we have modified the display of the UMAP graph to make it more comprehensible, i.e. by also showing the subpopulations identified using these markers, in agreement with the suggestions from reviewer #2.

Similarly, the peaks in Fig. 1b are not evenly spaced, particularly for low levels of fluorescence.

We can understand the concern of the reviewer about the spacing of the peaks. As the cells divide multiple times (>10), the peaks become less evident, e.g., Fig. 1b from day 6. For this revision, we tried to freshly stain $\gamma\delta$ T cells with CellTrace Violet at each timepoint to measure their proliferation during each timepoint intervals; 0-3, 3-6, 6-10, and 10-14. However, staining the cells during ongoing expansion impaired their proliferative capacity. To avoid confusion and inaccuracies, we removed the arrows pointing out individual peaks. The data shown in the revised Fig. 1b,d,k clearly show that the cells undergo significant numbers of divisions after the third day and that is the main message we want to convey.

As stated as a goal for the study, this report provides a cohesive framework for understanding functional responses of T cells. This report does provide new insight into changes in cell response over the course of activation and population expansion. However, this depth comes with a tradeoff in understanding how multiple types of cells affect $\gamma\delta$ T cell response. It is in fact surprising that the data of cell response is rather reproducible across multiple donors. The $\beta\gamma$ T cells represent a small fraction of PBMC, so the impact of additional cell types during activation should be discussed.

As the reviewer #3 pointed out, the functional experiments presented in our paper are performed on $\gamma\delta$ T cells isolated from the PBMC expansion culture. Therefore, in our assay, other cytotoxic cells such as conventional T cells and NK cells are absent and do not contribute to the killing.

A significant degree of reproducibility is expected between the $\gamma\delta$ T cells donors as PAgS are recognized by a large repertoire of V γ 9V δ 2 TCR which remains stable during the expansion despite the TCR affinity differences (e.g., Ravens et al., PNAS, 2020; doi.org/10.1073/pnas.19225881 or Fichtner et al., Journal of Leukocyte Biology, 2020; doi.org/10.1002/JLB.1MA0120-427RR).

Nonetheless, other cell types can play an important role for V γ 9V δ 2 T cell activation. Mainly, monocytes have been shown to present PAgS to the V γ 9V δ 2 T cells which participate to the immune reaction against multiple types of infections (Eberl et al., PLOS Pathogens, 2009; doi.org/10.1371/journal.ppat.1000308). Therefore, we have investigated if monocytes drive our *ex vivo* expansion of V γ 9V δ 2 T cells and their upregulation of granzyme B on day 3. Indeed, we could show that isolated monocytes are sufficient to activate V γ 9V δ 2 T cells which upregulated granzyme B to similar levels as in the traditional expansion in PBMCs (LINES 126-128). This data is now included in Extended Data Fig. 1j. The corresponding caption, the methods and the raw data excel files have been accordingly updated. We also included a brief contextualization of these results in the discussion section and compare our findings to *in vivo* models.

[REDACTED]

Fig. R2. Localization of rhPrf1 at sublytic concentration in the target cells. (a) rhPrf1 was added at different concentrations on A498 target cells including the dead cell staining SYTOX green for 5 h, to determine the rhPrf1 sublytic level (most of the cells survived at 60 ng/mL but not at 125 ng/mL). First row (1 min): imaging the cells just after adding the rhPrf1. Second row (5 h): the cells at the end of the assay. (b) Addition of rhPrf1 to A498 target cells at the indicated concentrations/incubation times prior to fixation, permeabilization, blocking and staining for perforin (clones δ G9 and B-D48), the early endosome marker EEA1 and the nucleus staining DAPI. The images were acquired by Airyscan microscopy and show the maximum intensity projections (of the multiple optical slices).

REVIEWERS' COMMENTS

Reviewer #1 (Remarks to the Author):

The authors have addressed all my concerns and suggestions as well as those from the other two Reviewers. As stated previously, this is an excellent study that provides important information on killing abilities of gamma delta T cells.

Reviewer #2 (Remarks to the Author):

The authors have provided a thoughtful and comprehensive response to previous reviews. This is a highly impactful study that advances our understanding of how less well-studied human lytic effector cells function using cutting-edge cell biological approaches. I have no further concerns.

Reviewer #3 (Remarks to the Author):

This revision addresses my comments from the original manuscript. It is an excellent study that sheds new light into the dynamics and cytotoxic function of $\gamma\delta$ T cells. I recommend acceptance of this work.

The new data on monocyte-based activation provides insight into natural variability in the interaction of $\gamma\delta$ T cells with a specific cellular component of the complex, physiological system. This sets a good foundation for understanding additional cell-cell interactions in that complex environment.

The modification to data presentation are appreciated. In particular, the reworking of the UMAP presentation is effective.